# PLUG-IN IMAGE QUALITY CONTROL FOR POSTERIOR DIFFUSION SUPER-RESOLUTION

## ABSTRACT

Diffusion-based super-resolution (SR) has shown remarkable progress, mainly through prior-guided approaches that require explicit degradation models or semantic priors. While posterior diffusion SR avoids these assumptions by directly learning from LR–HR pairs, it still suffers from numerical errors during sampling and lacks plug-in mechanisms for quality control. We introduce the first plug-in framework for posterior diffusion SR, enabling pretrained models to support controllable quality through marginal calibration, without retraining the core diffusion model. Our numerical analysis reveals that discretization errors are a key bottleneck in posterior SR. We prove that these errors can be equivalently expressed as gradients of KL divergence, unifying numerical error correction with image based classifier guidance. This provides a principled explanation of fidelity degradation and a new lens for posterior diffusion trajectories. In principle, these errors can be corrected to improve fidelity when reference supervision is available, offering a new theoretical understanding of posterior diffusion trajectories. In real-world SR, we further show that our image based guidance offers a controllable trade-off between fidelity and perception, delivering perceptual sharpness competitive with state-of-the-art prior-based models. Experiments confirm that our method consistently improves perceptual quality, while also validating the theoretical link between numerical errors and fidelity in posterior SR. These results position our work as a new direction for posterior diffusion models, bridging probabilistic analysis with practical deployment.

## 1 INTRODUCTION

Image super-resolution (SR) is a fundamental problem in low-level vision, aiming to recover high-resolution (HR) images from their low-resolution (LR) counterparts. Classical approaches include regression-based methods Mehta et al. (2023); Wang et al. (2023a); Lim et al. (2017); Liang et al. (2021) and adversarial training with GANs Ledig et al. (2017); Wang et al. (2021; 2018), but these methods still struggle with the intrinsic ill-posedness of SR.

Diffusion models Ho et al. (2020); Song et al. (2021b) have recently emerged as a powerful generative framework, leveraging a physics-driven probabilistic process that excels at reconstructing high-frequency details in ill-posed image restoration problems Saharia et al. (2021); Wang et al. (2023c); Li et al. (2022); Kawar et al. (2022). While diffusion models were initially criticized for their high computational cost, subsequent advances such as improved samplers Song et al. (2021a); Yue et al. (2023) and distillation-based acceleration Wu et al. (2024a); Dong et al. (2024); Chen et al. (2025; 2024c); Yue et al. (2025); Lin et al. (2024); Wu et al. (2024b); Yang et al. (2024) have reduced the sampling steps to single or few iterations, making runtime comparable to regression-based SR. Moreover, semantic priors such as CLIP have been employed to guide perceptual quality.

Despite these advances, most existing works adopt a *diffusion prior* paradigm: pretrained generative priors with explicit or learned degradation models Saharia et al. (2021); Li et al. (2022). However, diffusion priors require an accurate likelihood formulation of the degradation process, which is often intractable beyond simplistic assumptions such as Gaussian blur or bicubic downsampling. While higher-order solvers Lu et al. (2022; 2023); Zheng et al. (2023) mitigate discretization error in the generative trajectory, they remain dependent on assumed likelihoods that do not align well with real-world degradations, especially those introduced by optical lenses.

Table 1: Prior vs. Posterior Diffusion SR

| Aspect | Diffusion Prior SR | Posterior Diffusion SR |
|---|---|---|
| Training | Learn prior $p(x)$ + likelihood $p(y|x)$ | Directly learn $p(x|y)$ from LR–HR pairs |
| Assumption | Need explicit degradation model. (Bicubic, Gaussian) | Covered by the Dataset |
| Plug-in | CLIP Radford et al. (2021a) ControlNet Zhang et al. (2023) | Ours: the first plug-in model |
| Distillation | Actively used | Not Active |

In contrast, *posterior learning* approaches define the degradation process directly through LR-HR training pairs Cai et al. (2019); Wang et al. (2021); Bhat et al. (2021). This formulation bypasses the need for explicit likelihood modeling and naturally inherits the legacy of regression-based SR datasets. However, posterior diffusion models remain underexplored: unlike diffusion priors, no plug-in modules for image quality control have been established on pretrained posterior models.

From a numerical analysis perspective, diffusion samplers typically employ first-order forward discretization, which is computationally efficient but induces significant numerical error. Existing efforts Lu et al. (2022; 2023); Zheng et al. (2023) target discretization in diffusion priors, but posterior models have not been systematically analyzed in this regard. Notably, Yue et al. (2023) adopts a single-order solver, which remains susceptible to discretization error. For general image generation, Li et al. Li & van der Schaar (2024) analyzed cumulative error between forward and backward processes, but the role of discretization error in SR has yet to be explicitly studied.

**Our contribution.** The contributions of this work are summarized as follows:

- We propose the first *plug-in module* for posterior diffusion SR, enabling pretrained models to support controllable quality through marginal calibration, without retraining the core diffusion model.

- We theoretically derive that discretization errors in first-order posterior models can be corrected by incorporating second-order differential terms, providing a principled explanation of fidelity degradation.

- We further propose an *image-based classifier guidance formulation* for posterior diffusion and prove that it is mathematically equivalent to the second-order correction term, thereby unifying numerical analysis with guidance-based conditioning.

- While explicit fidelity correction requires HR references and is thus of limited use in blind SR, we show that the same formulation can be adapted in a sign-flipped manner to enhance perceptual quality in reference-free scenarios.

- This dual view—fidelity-oriented correction under supervision and perception-oriented control without supervision—establishes a unified framework for posterior SR.

- Extensive analysis and experiments validate that our approach bridges the gap between numerical error theory and real-world perceptual enhancement in posterior learning models.

## 2 RELATED WORKS

**Image Super-Resolution** Since the emergence of deep neural network-based image super-resolution Liang et al. (2021); Zhang et al. (2021); Dong et al. (2011); Gu et al. (2015); Zamir et al. (2022), significant advancements have been made in this field. Generative adversarial networks (GANs) Goodfellow et al. (2014) have played a crucial role in enhancing details in super-resolution outputs Wang et al. (2021; 2018). By leveraging adversarial learning, GANs improve texture sharpness and realism in super-resolved images. Recently, diffusion models Saharia et al. (2021); Li et al. (2022); Kawar et al. (2022); Yue et al. (2023); Wang et al. (2023c) have emerged as a powerful alternative for image super-resolution. Originally developed for novel image synthesis, these models have demonstrated exceptional capabilities in texture restoration and fine-detail reconstruction. Unlike GANs, diffusion models progressively refine images from noise, providing a more stable and controlled SR framework. Studies have shown that diffusion-based SR outperforms GAN-based approaches, particularly in preserving high-frequency details and reducing artifacts Rombach et al. (2022). Furthermore, developments in VQGAN Rombach et al. (2022) and stable diffusion models Rombach et al. (2022) have significantly influenced image synthesis Saharia et al. (2022), expanding beyond SR into text-conditioned image generation. Pretrained Stable Diffusion models have become widely adopted as diffusion priors, leading to a rapid increase in research on diffusion prior–based

super-resolution. With methods such as ControlNet Zhang et al. (2023) and CLIP Radford et al. (2021a), it has become possible to steer image generation or restoration toward desired semantic attributes, resulting in significant improvements in perceptual quality Lin et al. (2024); Wu et al. (2024b); Yang et al. (2024).

**Performance Enhancement** Diffusion models for image super-resolution face an inherent challenge of high computational overhead Saharia et al. (2021); Ho et al. (2020). Various efforts have been made to address this issue Song et al. (2021a); Lu et al. (2022; 2023); Zheng et al. (2023); Wu et al. (2024a); Wang et al. (2024). Starting with DDIM, subsequent works on DPM solvers have significantly reduced the number of iteration steps from thousands to just tens. Residual-based approaches in image synthesis Liu et al. (2024) and image super-resolution Yue et al. (2023) have demonstrated stable performance even with fewer than ten steps. More recently, techniques such as model distillation Wang et al. (2024) and single-step SR Wu et al. (2024a) have emerged, further pushing the boundaries of efficiency in diffusion-based SR.

**Error Estimation** Reducing the number of iteration steps inherently increases step intervals in finite difference methods, making diffusion models more susceptible to numerical errors Strang (2007); Peter & Eckhard (1992); Hochbruck & Ostermann (2005). Consequently, research has been conducted to analyze and mitigate numerical errors and their effects Li & van der Schaar (2024); Li et al. (2024). Trajectory analysis Chen et al. (2024b;a) is also effective for detailed analysis of numerical errors. In Li & van der Schaar (2024), the authors investigated errors in pretrained diffusion models by analyzing discrepancies between forward and reverse processes. They defined **modular errors** between these processes and extended their analysis to **cumulative errors** throughout the entire process. Similarly, Li et al. (2024) examined how iterative inference steps exacerbate exposure bias due to training-inference discrepancies and proposed a method to mitigate this issue without requiring DPM retraining.

**High Order Differential Equation Solver** Diffusion based super resolution initially began by learning the posterior distribution.Li et al. (2022); Saharia et al. (2021). However, the diffusion posterior sampling(DPS) approach has gained attention for its efficiency , as it enables sampling using a pre-trained prior without requiring additional training costs. Most existing higher-order ODE/SDE solvers (Heun, RK2/3, DPM-Solver-2/3, etc.) have been developed in the context of DDPM-like models, where the prior is learned and the sampling process is derived from a score-based or probability-flow ODE interpretation Song et al. (2021a); Karras et al. (2022); Lu et al. (2022; 2023); Zheng et al. (2023); Liu et al. (2022). These works typically apply numerical integration of a pre-trained prior distribution, thus enabling explicit higher-order schemes for fast sampling. However, compared to the DPS approach, which reuses a pre-trained model trained with a distribution based loss, the posterior learning using pixel-based loss is still considered advatangeous in terms of final image quality. Yue et al. (2023) Thus, our setting focuses on a **learned-posterior** approach rather than a learned-prior approach, where the solver is based on the first-order Euler-type update. The higher-order solvers introduced in prior-based diffusion works may not be readily applicable to the learned posterior method without re-architecting the model to fit a score-based ODE sampling paradigm.

# 3 PLUG-IN MODULE FOR POSTERIOR DIFFUSION MODEL

## 3.1 BACKGROUND FOR NUMERICAL ACCURACY

A Taylor series expansion can be written as

$$x(t + h) = x(t) + h\,\nabla x(t)\ +\ \frac{1}{2}\,h^2\,\nabla^2 x(t)\ +\ \mathcal{O}(h^3).$$

In a first-order difference scheme, we usually adopt

$$x(t + h) = x(t) + h\,\nabla x(t) + DE,$$

where $DE$ is the leading-order discretization error, whose dominant term is $\frac{1}{2}\,h^2\,\nabla^2 x(t)$. If we explicitly include this second-order term in the difference scheme, the accuracy increases. Most existing diffusion models, however, only consider a first-order discretization, whereas in this paper, we adopt a second-order discretization that accounts for $\frac{1}{2}\,h^2\,\nabla^2 x(t)$. Higher-order discretization methods and their errors have also been analyzed in Lu et al. (2022; 2023); Zheng et al. (2023). While

conventional formulas rely on starting the restoration from white Gaussian noise (thus requiring many diffusion steps), recent works on one-step or few-step diffusion models employ large step intervals, which can cause large discretization errors. Therefore, it becomes necessary to correct such discretization errors.

**DPM-Solver Revisited.**  Consider a random variable $x_0$ sampled from $q_0(x_0)$. The forward SDE on the interval $[0, T]$ is

$$d\mathbf{x}_t = f(t)\,\mathbf{x}_t\,dt + g(t)\,d\mathbf{w}_t, \quad \mathbf{x}_0 \sim q_0(\mathbf{x}_0),$$

where $\mathbf{w}_t \in \mathbb{R}^D$ is a standard Wiener process. Song et al. (2021b) show that its reverse process from $T$ to 0, given the marginal $q_T(x_T)$, is

$$d\mathbf{x}_t = \left[ f(t)\,\mathbf{x}_t - g^2(t)\,\nabla_{\mathbf{x}} \log p_t(\mathbf{x}_t) \right] dt + g(t)\,d\bar{\mathbf{w}}_t, \quad \mathbf{x}_T \sim p_T(\mathbf{x}_T),$$

where $\bar{\mathbf{w}}_t$ is a standard Wiener process in reverse time. From the above, Lu et al. (2022; 2023) transform it into a probability flow ODE for faster sampling:

$$d\mathbf{x}_t = \left[ f(t)\,\mathbf{x}_t - \tfrac{1}{2}g^2(t)\,\nabla_{\mathbf{x}} \log p_t(\mathbf{x}_t) \right] dt. \tag{1}$$

To estimate the score function $\nabla_{\mathbf{x}} \log q_t(x_t)$, DPMs use a neural network $\epsilon_\theta(x_t, t)$. The parameter $\theta$ is optimized by minimizing

$$\mathcal{L}(\theta) = \int_0^T \mathbb{E}_{q_t(\mathbf{x}_t)} \big\| \boldsymbol{\epsilon}_\theta(\mathbf{x}_t, t) + \nabla_{\mathbf{x}} \log p_t(\mathbf{x}_t) \big\|_2^2 \, dt,$$

yielding

$$d\mathbf{x}_t = f(t)\,\mathbf{x}_t + g^2(t)\,\boldsymbol{\epsilon}_\theta(\mathbf{x}_t, t)\, dt.$$

We can generate samples by numerically solving this ODE from $T$ down to 0. Given an initial value $x_s$ at time $s > 0$, Lu et al. (2022) show that the solution $x_t$ for $t \in [0, s]$ can be written in integral form as

$$\mathbf{x}_t = \frac{\alpha_t}{\alpha_s}\,\mathbf{x}_s + \alpha_t \int_{\lambda_s}^{\lambda_t} e^{-\lambda}\,\boldsymbol{\epsilon}_\theta(\mathbf{x}_\lambda)\, d\lambda, \tag{2}$$

which in discrete form approximates

$$\mathbf{x}_t = \frac{\alpha_t}{\alpha_s}\,\mathbf{x}_s - \alpha_t \sum_{n=0}^{k-1} \boldsymbol{\epsilon}_\theta^{(n)}(\mathbf{x}_{\lambda_t}) \int_{\lambda_s}^{\lambda_t} F(\lambda)\, d\lambda, \tag{3}$$

where $F(\lambda)$ collects certain integral factors (see Lu et al. (2022)), and $\epsilon_\theta^{(n)}(\cdot)$ is the $n$-th order derivative for $n \leq k-1$. For $k \geq 2$, they approximate these derivatives using intermediate points (Runge–Kutta). Although evaluating the integral more finely can reduce the discretization error, in practice it is typically done via finite-difference approaches like Runge–Kutta Hochbruck & Ostermann (2005). In our paper, however, we propose directly computing the second-order derivatives by taking gradients of the neural network model.

### 3.2  BACKGROUND FOR INVERSE PROBLEM SOLVER

Super-resolution (SR) is an ill-posed inverse problem, where the goal is to recover the high-resolution (HR) image $\mathbf{x}$ from its degraded low-resolution (LR) observation $\mathbf{y}$. The forward measurement process can be expressed as $\mathbf{y} = \boldsymbol{A}(\mathbf{x}) + \mathbf{n}, \quad \mathbf{y}, \mathbf{n} \in \mathbb{R}^n, \ \mathbf{x} \in \mathbb{R}^d$ where $\boldsymbol{A}(\cdot) : \mathbb{R}^d \to \mathbb{R}^n$ is the degradation operator (e.g., bicubic downsampling, blur, or camera pipeline), and $\mathbf{n}$ denotes measurement noise. From a Bayesian perspective, the posterior distribution is given by

$$p(\mathbf{x}|\mathbf{y}) = p(\mathbf{y}|\mathbf{x})\,p(\mathbf{x})/p(\mathbf{y})$$

where $p(\mathbf{x})$ the prior over natural images, $p(\mathbf{y}|\mathbf{x})$ the likelihood for degradation, $p(\mathbf{x}|\mathbf{y})$ the posterior. In **Diffusion Prior** methods Saharia et al. (2021); Li et al. (2022); Wang et al. (2023c); Wu et al. (2024b), the diffusion model serves as the image prior $p(\mathbf{x})$. The likelihood $p(\mathbf{y}|\mathbf{x})$ is typically

assumed to be a Gaussian degradation model: $p(\mathbf{y}|\mathbf{x}) = \mathcal{N}(\mathbf{y}; A(\mathbf{x}), \sigma^2 I)$. Conditioning the diffusion process on $\mathbf{y}$ yields a posterior-guided reverse ODE:

$$d\mathbf{x}_t = \left[ f(t)\,\mathbf{x}_t - \tfrac{1}{2}g^2(t)\,\nabla_\mathbf{x} \log p_t(\mathbf{x}_t|\mathbf{y}) \right] dt, \tag{4}$$

where the score term $\nabla_\mathbf{x} \log p_t(\mathbf{x}_t|\mathbf{y})$ incorporates the assumed likelihood. Thus, the accuracy of prior-based SR crucially depends on the correctness of the degradation model.

**Posterior learning** approaches Yue et al. (2023) directly learn the conditional distribution $p(\mathbf{x}|\mathbf{y})$ from paired LR–HR data, bypassing the need for an explicit likelihood model. In this case, the degradation process $A(\cdot)$ does not need to be analytically specified; it is implicitly encoded through the training dataset. The training objective is to maximize the conditional log-likelihood: $\mathcal{L}(\theta) = \mathbb{E}_{(\mathbf{x},\mathbf{y})\sim\mathcal{D}} \left[ \log p_\theta(\mathbf{x}|\mathbf{y}) \right]$. This makes posterior approaches attractive for real-world SR, where the degradation is complex or unknown. However, posterior SR is sensitive to discretization errors during sampling and currently lacks plug-in quality control mechanisms, which are more natural in prior-based frameworks.

## 3.3 THEORETICAL FOUNDATION

Error estimation in diffusion models has been addressed by Lu et al. (2022; 2023); Zheng et al. (2023); Li & van der Schaar (2024); Li et al. (2024). Most prior work focuses on measuring the difference between forward and reverse processes or errors from sampling intervals but their restoration has been less studied. Our interest lies in the discretization errors of finite-difference methods Strang (2007) and SDEs Peter & Eckhard (1992). Below, we define how we measure discretization error and Kullback–Leibler(KL)-based divergence error in our diffusion model for image restoration.

**Assumption 1. (Definition of Error in the Diffusion Path)** In Li & van der Schaar (2024), they defined the error in path of diffusion trajectory. The modular error $\mathbf{E}_t^{\mathrm{modular}}$ measures the accuracy that every module maps its input to the output. $\mathbf{E}_t^{\mathrm{modular}} = \mathbb{E}\left[ D_{\mathrm{KL}}\left( p_\theta(\mathbf{x}_{t-1} \mid \mathbf{x}_t) \,\|\, q(\mathbf{x}_{t-1} \mid \mathbf{x}_t) \right) \right]$ The cumulative error $\mathbf{E}_t^{\mathrm{cumu}}$ measures the amount of error which are accumulated for sequentially running the first $T-1$ denoising modules. $\mathbf{E}_t^{\mathrm{cumu}} = D_{\mathrm{KL}}\left( p_\theta(\mathbf{x}_{t-1}) \,\|\, q(\mathbf{x}_{t-1}) \right)$ Here $p_\theta(\cdot)$ and $q(\cdot)$ denote the model distribution in the backward process and reference distribution in the forward process, following the notation of Li & van der Schaar (2024).

**Assumption 2 (Isotropic Variance Schedule).** As in ResShift Yue et al. (2023), we assume that the forward process uses a time–dependent scalar noise schedule $\{\sigma_t^2\}_{t=1}^T$ with spatially isotropic Gaussian noise at each diffusion step $t$ by $q(x_t \mid x_{t-1}, e_0) = \mathcal{N}(x_{t-1} + \alpha_t e_0, \ \sigma_t^2 I)$, where $\sigma_t^2 = \kappa^2 \alpha_t, e_0 = y_0 - x_0$ We also approximate the marginal at step $t$ by $q(x_t \mid x_0, y_0) \approx \mathcal{N}(\mu_t(x_0, y_0), \ \sigma_t^2 I)$. Thus, the variance is spatially constant within each timestep (isotropic covariance $\sigma_t^2 I$), while its magnitude can vary with $t$ according to the predefined schedule, exactly as in ResShift.

**Definition 3.2.1 (Numerical Error as Second-order derivative)** We define the *numerical error* (DE) of the first-order derivative solution as

$$\mathbf{x}_t^{\mathrm{Exact}} - \mathbf{x}_t^{\mathrm{Euler(1st)}} \approx \nabla_t\left[ f(t)\,\mathbf{x}_t - \tfrac{1}{2}g^2(t)\,\nabla_\mathbf{x} \log p_t(\mathbf{x}_t) \right].$$

It is the local truncation error of numerically integrating the backward ODE by Euler's method.

**Lemma 3.2.1 (Gaussian KL gradient)** For two normal distributions $\mathcal{P} : \mathcal{N}(\mathbf{x}_t, \sigma_1^2), \mathcal{Q} : \mathcal{N}(\boldsymbol{\mu}_t, \sigma_2^2)$, this shows that the difference between two Gaussian distributions can be expressed as a distance between their means.

$$\nabla_\mathbf{x} D_{\mathrm{KL}}\left( \mathcal{P}||\mathcal{Q} \right) \approx \frac{(\mathbf{x}_t - \boldsymbol{\mu}_t)}{\sigma_2^2}.$$

**Proposition 3.2.1 (Local linearization of Numerical Error).** Let $\sigma_t^2$ be time dependent and a predefined constant over the spatial dimensions. Then, the gradient with respect to $t$ of the modified score function from Definition 3.2.1 satisfies

$$\nabla_t\left[ f(t)\,\mathbf{x}_t - \tfrac{1}{2}g^2(t)\,\nabla_\mathbf{x} \log p_t(\mathbf{x}_t) \right] \approx A_t \cdot \frac{(\mathbf{x}_t - \boldsymbol{\mu}_t)}{\sigma_t^2} + B_t.$$

where $A_t, B_t \in \mathbb{R}$ are step-dependent coefficients to minimize discretization error. With constant variance, the numerical error can be found as a linear equation of the distance of two distributions.

**Proposition 3.2.2 (Approximation Relationship between Numerical Error and KL Gradient).** For a given step $t$, the difference between the exact solution $\mathbf{x}^{\text{Exact}}$ and the Euler discretization $\mathbf{x}^{\text{Euler}}$ can be expressed as

$$\mathbf{x}_t^{\text{Exact}} - \mathbf{x}_t^{\text{Euler}} \approx A_t \cdot \nabla_{\mathbf{x}} D_{\text{KL}}(\mathcal{P} \,\|\, \mathcal{Q}) + B_t.$$

where $\mathcal{P}$ is discrete distribution, $\mathcal{Q}$ is continuous exact distribution. Here, $A_t, B_t$ can be derived exactly through linear regression between latent spaces of SR and HR . Intuitively, this tell us that discretization error at each step behaves like the KL gradient between the continuous and discrete processes.

**Proposition 3.2.3 (Alternative KL Usage Under Lipschitz Continuity).** Under a suitable Lipschitz continuity assumption, the KL divergence term $D_{\text{KL}}(p_t \,\|\, q_{\text{HR}})$ may be replaced with $D_{\text{KL}}(p_t \,\|\, q_{\text{LR}})$, and moreover,

$$\nabla_{\mathbf{x}} D_{\text{KL}}(p_t \,\|\, q_{\text{HR}}) \;\approx\; \nabla_{\mathbf{x}} D_{\text{KL}}(p_t \,\|\, q_{\text{LR}}).$$

**Discussion.** The detailed proofs of the lemma, proposition are provided in the supplemental material. In summary, starting from Assumption 2, we derive Proposition 3.2.2, which establishes that the numerical error—characterized by a second-order derivative—can be formulated as the Kullback–Leibler (KL) divergence between distribution from the discrete process and the continuous process. Assumption 1 defined that cumulative error in a path is defined as divergence of the forward and backward process. Likewise, we assume that exact distribution is substituted with the distribution of the forward process and Euler distribution is with the distribution of the backward process. Proposition 3.2.2 define the error in a single step depends linearly on the gradient of KL divergence of the forward and backward process.

The distribution of the continuous process becomes the reference of the discrete process, which can be substituted with the forward process of HR image because it has better accuracy than the reverse process starting from LR image. Specifically, in the context of a conditional diffusion framework, the backward process is initialized from the low-resolution (LR) image, while the forward process originates from the high-resolution (HR) image.

Consequently, we demonstrate that the second-order numerical discretization error exhibits a linear dependency on the KL divergence between the latent representations of the HR and SR images and the SR and LR images.

$$\Delta = \mathbf{x}_t^{\text{Exact}} - \mathbf{x}_t^{\text{Euler}} \approx \nabla_{\mathbf{x}} D_{\text{KL}}\left(p_{\text{SR}} \,\|\, p_{\text{HR}}\right) \approx \nabla_{\mathbf{x}} D_{\text{KL}}\left(p_{\text{SR}} \,\|\, p_{\text{LR}}\right)$$

Thus, although the original discretization error is theoretically defined using $D_{\text{KL}}(p_{\text{SR}} \,\|\, p_{\text{HR}})$, real-world SR settings do not provide access to HR references. By leveraging Proposition 3.2.3, we replace this term with $D_{\text{KL}}(p_{\text{SR}} \,\|\, p_{\text{LR}})$, which serves as a practical surrogate for estimating and correcting numerical error. This surrogate formulation enables our method to perform error correction in both reference and non-reference regions, as detailed in the next section.

In the preceding paragraphs, we defined how the second-order numerical error can be evaluated and corrected. The next step is to formulate the theoretical framework for appyling these equations to the existing posterior model. From Yue et al. (2023), the reverse process for estimating the posterior distribution $p(\mathbf{x}_0 \mid \mathbf{y}_0)$ is defined by:

$$p(\mathbf{x}_0 \mid \mathbf{y}_0) = \int p(\mathbf{x}_T \mid \mathbf{y}_0) \prod_{t=1}^{T} p_\theta(\mathbf{x}_{t-1} \mid \mathbf{x}_t, \mathbf{y}_0) \, \mathrm{d}\mathbf{x}_{1:T}. \tag{5}$$

where $x_0$ is the data distribution and $y_0$ is a measurement or LR. From above equation, we need to incorporate numerical error correction term from Proposition 3.2.2 into the above equation. For this purpose, we provide an additional LR image as a condition for inference to the existing conditional diffusion model. To achieve this, we extend the ResShift formula with the explicit guided diffusion approach from Dhariwal & Nichol (2021). In Eq. 5, $x_0$ and $y_0$ represent the HR and LR images, respectively.

$$p(\mathbf{x}_0 \mid \mathbf{y}_0, \mathbf{c}) = \int p(\mathbf{x}_T \mid \mathbf{y}_0, \mathbf{c}) \prod_{t=1}^{T} p_\theta(\mathbf{x}_{t-1} \mid \mathbf{x}_t, \mathbf{y}_0, \mathbf{c}) \, \mathrm{d}\mathbf{x}_{1:T}, \tag{6}$$

where $\mathbf{c}$ is an additional LR condition.

**Proposition 3.3.1 (Guidance for LR-based Conditioning)**    The target distribution $p(\mathbf{x}_0 \mid \mathbf{y}_0, \mathbf{c})$ with LR guidance is given by

$$\mathcal{N}\Big(\boldsymbol{\mu} \; + \; \Sigma \, \nabla p_\theta(\mathbf{c} \mid \mathbf{x}_t, \mathbf{y}_0, t), \; \Sigma\Big), \tag{7}$$

where $\mathbf{c}$ indicates the additional image guidance.

**Remark 3.3.1 (Implementation and Partial Guidance).**    From Proposition 3.2.2 and Proposition 3.3.1, one obtains:

$$\nabla_{\mathbf{x}} \log p\big(\mathbf{c} \mid \mathbf{x}_t, \mathbf{y}_0\big) = \nabla_{\mathbf{x}} D_{\mathrm{KL}}\big(p_\theta(\mathbf{x}_t) \, \| \, p_\theta(\mathbf{c})\big), \tag{8}$$

for an image-based guidance approach. This remark connects numerical theory to practice: numerical correction and classifier guidance have the equivalence.

**Discussion.**    With Remark 3.3.1, we derive a theoretical basis for using an input image as an additional condition alongside the conventional LR input. In practical scenarios, the conventional condition is passed through a noise-added pipeline as part of the diffusion process, while the newly introduced condition is incorporated through a noise-free path. This dual-conditioning strategy allows us to correct the discretization error accumulated during few-step sampling. Please note that the numerical error correction term is equivalent to our image-based classifier guidance term, up to a constant. This demonstrates that numerical error correction can be realized through image-based classifier guidance, enabling a plug-in mechanism without retraining the core diffusion model. More detailed description about practical usage will be introduced in the next section.

### 3.4 Image Control via Image-Based Guidance

Based on previous formulations, the second-order numerical error can be represented by the gradient of the KL divergence between the SR and HR distributions. This implies that the second-order numerical error can be linearly regressed using the gradient of the KL divergence, thereby making numerical error restoration theoretically feasible. Simultaneously, from Remark 3.3.1, we defined a method how to supply additional image to the diffusion pipeline as the classifier condition where it is also the gradient of the KL divergence of latent vector and classifier condition. Therefore, if we want to restore numerical error of the posterior model with HR image or LR image, we just need to supply HR image or LR image as classifier condition.

**Reference Region** The following procedure demonstrates how to estimate the constants $A_t$ and $B_t$ by measuring the difference in the latent space between SR and HR within the diffusion model and regressing it against the divergence between the two distributions.

$$\Delta = \mu_t(p_{\mathrm{HR}}) - \mu_t(p_{\mathrm{SR}}) \tag{9}$$

$$A_t, B_t = \arg \min_{a_t \in \mathbb{R}, \; b_t \in \mathbb{R}} \|\Delta - (a_t \cdot \nabla_{\mathbf{x}} D_{\mathrm{KL}}(p_{\mathrm{SR}} \, \| \, p_{\mathrm{HR}}) + b_t)\|_2^2 \tag{10}$$

$$\mu_{\mathrm{new}}(p_{\mathrm{SR}}) = \mu_{\mathrm{old}}(p_{\mathrm{SR}}) + B_t + A_t \cdot \nabla_{\mathbf{x}} D_{\mathrm{KL}}(p_{\mathrm{SR}} \, \| \, p_{\mathrm{HR}}) \tag{11}$$

where $A_t, B_t \in \mathbb{R}$. $\nabla_{\mathbf{x}} D_{\mathrm{KL}}(p_{\mathrm{SR}} \, \| \, p_{\mathrm{HR}})$ follows the trajectory of the surface between SR and HR.Chen et al. (2024b) Therefore, it can restore truncation error in Proposition 3.2.2. This will be validated in Figure 1, 10.

**Non-Reference Region** In real-world blind super-resolution, it is not feasible to access HR references, and thus the difference between HR and SR (Eq. 9) as well as the corresponding trajectory (Eq. 10) are unavailable. To address this, we propose to estimate the parameters $A_t$ and $B_t$ from regions where LR–HR pairs exist, and then apply them to LR-only regions. Since Eq. 10 does not exist in such settings, we instead utilize $\nabla_{\mathbf{x}} D_{\mathrm{KL}}(p_{\mathrm{SR}}|p_{\mathrm{LR}})$ following Proposition 3.2.3. While Eq. 10 corresponds to the gradient at each time step along the SR–HR trajectory, if the LR–SR trajectory resided in the extrapolated space of the SR–HR trajectory, one could substitute $D_{\mathrm{KL}}(p_{\mathrm{SR}}|p_{\mathrm{HR}})$ with $D_{\mathrm{KL}}(p_{\mathrm{SR}}|p_{\mathrm{LR}})$ and compute $A_t$ and $B_t$ to construct the correction term. Our validation experiments in Fig. 5 confirmed that they are not well aligned, particularly in the pretrained model of Yue et al. (2023). Instead, we propose another calibration scheme that approximates numerical errors using $D_{\mathrm{KL}}(p_{\mathrm{SR}}|p_{\mathrm{LR}})$. This scheme is summarized in Algorithm 1 and Algorithm 2.

**Algorithm 1** Calibrate Guidance Scale

**Require:** A set of LR–HR pairs $(\mathbf{LR}_i, \mathbf{HR}_i)$ model parameters $\theta$, candidate scale set $\{\alpha_t\}$
**Ensure:** A chosen guidance scale $\alpha_t^*$.
1: **for** $i = 1$ to $n$ **do**
2:      $\mathcal{Z}_T^{(\text{HR})} \leftarrow \text{ForwardODE}(p_{\text{HR}_i})$
3:      $x_T \leftarrow \mathbf{LR}_i$
4:      **for** $t = T$ down to 1 **do**
5:          $(\mu, \Sigma) \leftarrow \mu_\theta(x_t),\ \Sigma_\theta(x_t)$
6:          $\Delta = \mu_\theta(\mathcal{Z}_T^{(\text{HR})}) - \mu$
7:          $G_{\text{LR}} = \nabla_{x_t}\, D_{\text{KL}}\big(p_\theta(x_t)\,\|\,p_{\text{LR}_i}\big)$
8:          $L_i(\alpha_t) = \big\|\Delta\big\| - \big\|\alpha_t \cdot G_{\text{LR}}\big\|$
9:      **end for**
10: **end for**

11: $\alpha_t^* = \underset{\alpha_t}{\arg\min} \sum_{i=1}^{n} L_i(\alpha_t).$

12: **return** $\alpha_t^*$

**Algorithm 2** Inference with LR Guidance

**Require:** LR image (or condition) $c$, guidance scale $\alpha_t^*$, diffusion model parameters $\theta$,
**Ensure:** Reconstructed SR image $x_0$.
1: $x_T \leftarrow \mathbf{LR}_i$
2: **for** $t = T$ down to 1 **do**
3:      $(\mu, \Sigma) \leftarrow \mu_\theta(x_t),\ \Sigma_\theta(x_t)$
4:      $x_{t-1} \leftarrow$
5:      $\mathcal{N}\big(\mu + \alpha_t^* \Sigma \nabla_{x_t}\, D_{\text{KL}}\big(p_\theta(x_t)\,\|\,p_{\text{LR}_i}\big)$
6: **end for**
7: **return** $x_0$

The process of determining parameters $A_t$ and $B_t$ can be regarded as a calibration step, where multiple LR–HR image pairs are used to identify numerical errors and extract the corresponding parameters, which are then provided during inference. Linear regression over multiple images failed to yield optimized parameters for both $A_t$ and $B_t$. Consequently, we abandon the estimation of $B_t$ and instead focus solely on finding $A_t$ which is a vector with the size of diffusion step, which minimizes numerical errors across multiple images and is supplied during inference. More details about how to derive algorithm 1 and algorithm 2 will be provided in Fig. 4 and in section 6.

## 4 EXPERIMENT

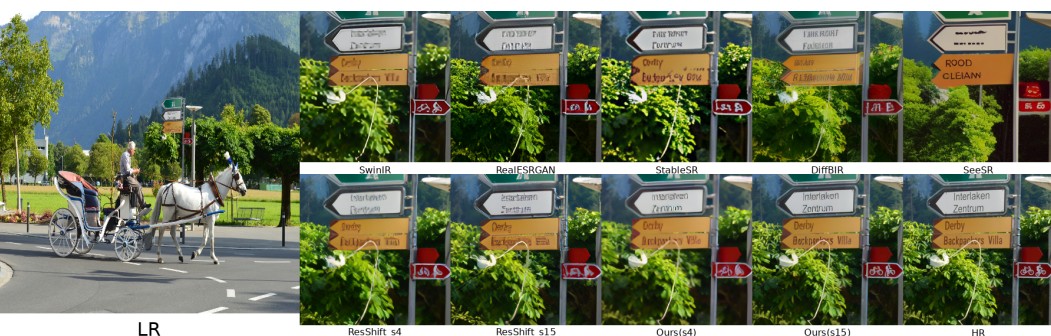

Figure 1: **Reference Region**. Ours (s4) and Ours (s15) represent ResShift (s4 and s15) with the proposed correction module in Eq. 11. In the region with reference overlap, the KL divergence gradient between HR and SR is used to restore the numerical error. This demonstrates that the proposed method effectively corrects the error in the overlapped region. After numerical error correction, architectural features such as the text on travel signboards were successfully restored. In particular, the s15 model achieved restoration results nearly indistinguishable from the HR image because its truncation error is already lower than that of the s4 model.

**Testing Environment** To validate our theoretical framework, we conducted experiments using the DIV2K validation dataset Agustsson & Timofte (2017) , the RealSR dataset Cai et al. (2019), and Flickr30k Dataset Young et al. (2014). Since datasets contain images of varying sizes, we cropped $128 \times 128$ center region and further divided them into four $128 \times 128$ patches. We used 100 images for DIV2K and RealSR for generation of quantitative table. Calibration of Guidance scale is performed over Imagenet Dataset Deng et al. (2009) which is used in training the pretrained model of Yue et al. (2023). We conducted all experiments on Nvidia H100 machine.

**Metrics for comparison** For a comprehensive image quality assessment, we employ both full-reference and no-reference metrics. Reference-based fidelity measures: PSNR and SSIM Wang et al. (2004) / Reference-based perceptual quality measures: LPIPS Zhang et al. (2018) and DISTS Ding

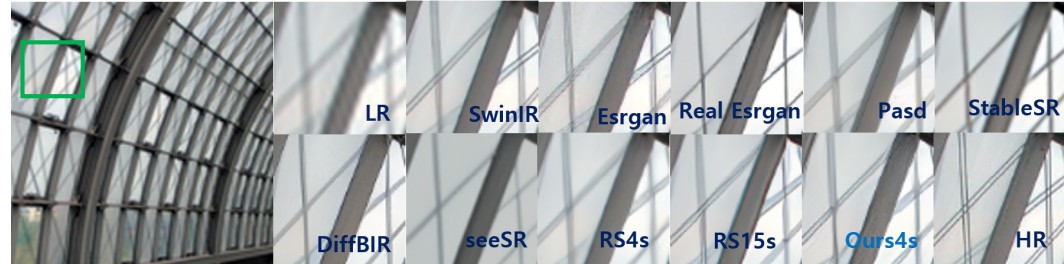

Figure 2: **Non-Reference Region** Qualitative comparisons in the non-reference region. Ours shows perceptual quality enhancement in the non-overlapped region with the reference according to the Remark 3.3.1

et al. (2020) / No-reference image quality measures: NIQE Zhang et al. (2015), MANIQA Yang et al. (2022), and CLIPIQA Wang et al. (2023b) / Image distribution-based metric: FID Heusel et al. (2017), which evaluates the distance between restored images and ground truth.

To validate the effectiveness of our proposed method, we conducted quantitative and qualitative comparisons against various state-of-the-art (SOTA) methods, including: SwinIR Liang et al. (2021), Real-ESRGAN Wang et al. (2021), StableSR Wang et al. (2023c), ResShift Yue et al. (2023), PASD Yang et al. (2024), DiffBIR Lin et al. (2024), SinSR Wang et al. (2024), OseDiff Wu et al. (2024a), InvSR Yue et al. (2025), SeeSR Wu et al. (2024b). Our comparative analysis includes regression-based, GAN-based, and diffusion-based super-resolution methods. In Table 2, the best and second-best results are highlighted in red and blue, respectively, while regression- and GAN-based methods were excluded from this ranking. To the best of our knowledge, ResShift Yue et al. (2023) is currently the only posterior-based super-resolutin model. Thereby, we adopt ResShift Yue et al. (2023) as our backbone framework. Other methods based on pretrained diffusion prior such as Rombach et al. (2022) are PASD Yang et al. (2024), StableSR Wang et al. (2023c), and DiffBIR Lin et al. (2024). They incorporate additional neural networks(ControlNet Zhang et al. (2023) and CLIPRadford et al. (2021b)) into the diffusion prior model. This could lead to overlapping effects between our theoretical enhancement and their inherent framework improvements.

**Qualitative Analysis.** Figures 1 and 2 compare our plug-in module on reference and non-reference regions. Figure 1 illustrates the reference-based case in which parameters of Eq. 10 are estimated using the HR latent representation. With available HR information, the corrections are highly accurate, enabling precise restoration of fine structures that competing SR models fail to reconstruct (e.g., text on signboards). Although HR is not available in real-world blind SR, *this controlled experiment verifies the correctness of our formulation*. Practical usage of this concept for real-world SR is discussed in Section 9.

Figure 2 presents real-world SR obtained by Algorithms 1 and 2, where parameters are evaluated from training samples. The results show that our method effectively restores the perceptual quality of posterior models. Our plug-in module is integrated into the ResShift-4s or ResShift-15s framework. Due to discretization error, the 4-step model inherently produces lower-quality outputs than the 15-step version. In the HR ground truth, all wires follow a dual-wire structure. Among the SR results, only SwinIR, ResShift-15s, and our method successfully reconstruct this structure. As expected, the 15-step model suffers less from discretization error than the 4-step model when both are properly trained. *Remarkably, despite operating with only four steps, our method succeeds in reconstructing the dual-wire structure.*

**Quantitative Analysis.** From Table 2, our algorithm achieves state-of-the-art performance in non-reference metrics such as MANIQA, CLIPIQA, and FID, and also improves perception-oriented reference metrics such as LPIPS, while minimizing degradation in reference-based fidelity metrics. Our model in Table 2 uses negatively signed scale parameters to enhance perceptual quality. Table 3 further analyzes performance in reference/non-reference regions and across positive/negative scale parameter settings. The implementation details associated with Table 3 are provided in Section 6.1. The `ours(hr,+)` variant operates in reference regions and therefore successfully restores high-frequency HR details, achieving fidelity scores close to the upper bound of the 4-step baseline model. The `ours(hr,-)` variant represents the opposite correction direction; although some outputs in Figure 4 may appear visually plausible, the negative sign relies on information that does not exist in the LR input, causing fidelity to collapse and producing unrealistic results. As discussed in Yue

et al. (2023), fidelity and perceptual quality exhibit a trade-off in most SR models. Our results in Table 3 confirm this behavior for both ours(lr,+) and ours(lr,-). With positive sign parameters, ours(s4,LR,+) improves fidelity while slightly lowering perceptual quality. With negative parameters, the opposite behavior emerges. This reflects a fundamental characteristic of diffusion-based SR. As shown in Tables 2 and 3, metric rankings follow different orders:

$$\text{PSNR/SSIM:} \quad \text{SR} \rightarrow \text{bicubic(LR)} \rightarrow \text{HR},$$
$$\text{LPIPS/DISTS/NIQE:} \quad \text{bicubic(LR)} \rightarrow \text{SR} \rightarrow \text{HR},$$
$$\text{MANIQA/CLIPIQA:} \quad \text{bicubic(LR)} \rightarrow \text{HR} \rightarrow \text{SR}.$$

This behavior is tied to diffusion models' strong denoising properties. Since PSNR/SSIM penalize deviations in low-level noise statistics, bicubic(LR) —which shares noise statistics with HR—often scores surprisingly well. Diffusion models, being powerful denoisers, are disadvantaged in these metrics despite superior visual quality. Figures 12 visually confirm this phenomenon. As diffusion SR aims to make objects appear sharper and more recognizable, recent trends prioritize non-reference metrics. We follow this trend, using negatively signed parameters to enhance perceptual quality in real-world SR. The perceptual improvement is evident in Table 2, where our parameter-optimized plug-in module achieves state-of-the-art performance. In our model, maximizing a single metric is trivial; achieving balanced performance across all metrics is substantially more challenging, which is why accurate parameter estimation is critical.

In our formulation, the calibrated parameters obtained via loss evaluation naturally push the model's mean toward the HR direction. Since HR outperforms SR in reference-based metrics, positive-sign correction improves these metrics for both ours(hr,+) and ours(lr,+), as shown in Table 3. However, in perceptual metrics, diffusion SR often outperforms HR; thus, moving closer to HR degrades perceptual scores. This is visible in Figure 4 for the HR-based case and Figure 13 for the LR-based case. With negative-sign parameters, the correction moves away from HR—lowering fidelity but increasing perceptual quality. Thus, our model follows the perception–distortion trade-off Yue et al. (2023), but allows explicit and continuous control over the balance. Nevertheless, excessively large negative parameters produce severe artifacts (e.g., grain), as seen in Figure 13. Although negative-signed parameters do not minimize the loss, their usable range is still governed by the loss-associated structure of our formulation. This relationship is demonstrated in Table 4. Row [3] corresponds to positively calibrated optimal parameters for error correction. For a 4-step model, three parameters are estimated, and for a 15-step model, fourteen parameters (since the variance is zero at $t = T$). The table reports the error between the absolute summation of the old and updated model means. Row [3] shows consistent error reduction for properly calibrated parameters, but insufficiently calibrated positive parameters such as row [4] can increase error, as seen at step 2. For negative-sign parameters, error increases but remains bounded, as shown in row [2]; the corresponding images (third in Figure 13) remain visually acceptable. Extremely large negative parameters such as row [1], however, cause sharp error escalation and produce the artifacts shown in the fourth and fifth images of Figure 13. Thus, in perceptual enhancement, the positive-sign optimal parameter acts as an absolute upper bound of the negative-sign parameters. ***The procedure for obtaining optimal scale parameters is detailed in Fig. 6 and 7(b). The underlying mechanism of the sign-flip is available in Section 10***.

## 5    CONCLUSION

In this work, we introduced a plug-in module for posterior learning-based super-resolution models. From a theoretical perspective, we derived that discretization errors inherent in first-order ODE formulations can be corrected by incorporating second-order differential terms. We further showed that this correction can be formulated within the conditional diffusion framework, and that the resulting expression naturally coincides with the structure of image-based classifier guidance. This provides a new theoretical understanding of posterior diffusion trajectories.

From a practical perspective, we demonstrated that applying the correction term can reduce numerical errors in pretrained models with marginal computation for parameter calibration. While fidelity enhancement through this correction requires HR supervision and is thus limited in blind SR scenarios, we proposed instead to exploit LR-based classifier guidance in a sign-flipped manner to enhance perceptual quality. This design enables posterior SR models to achieve perceptual results competitive with state-of-the-art methods, thereby bridging theoretical insight with practical applicability.

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
