CONTENTS

Table 2: Quantitative comparison with contemporary state-of-the-art image super resolution algorithms for non-overlapped region. 's' denotes the number of diffusion steps. The best and second-best results of each metric are highlighted in red and blue, respectively.

| Datasets | Methods | PSNR↑ | SSIM↑ | LPIPS↓ | DISTS↓ | NIQE↓ | MANIQA↑ | CLIPIQA↑ | FID↓ |
|---|---|---|---|---|---|---|---|---|---|
| RealSR | HR | inf | 1.00000 | 0.00000 | 0.00000 | 4.46300 | 0.48905 | 0.63147 | 0.00005 |
| | Bicubic | 25.93227 | 0.757129 | 0.28747 | 0.238012 | 6.76492 | 0.40826 | 0.50845 | 68.46397 |
| | SwinIR | 26.401153 | 0.822698 | 0.198882 | 0.212861 | 5.533595 | 0.482269 | 0.520561 | 34.319923 |
| | Realesrgan | 24.824283 | 0.690881 | 0.228614 | 0.211228 | 5.554380 | 0.515528 | 0.602108 | 68.559638 |
| | StableSR(s200) | 23.141011 | 0.700490 | 0.251224 | 0.218401 | 5.993226 | 0.445875 | 0.619377 | 66.324215 |
| | DiffBIR(s50) | 24.044400 | 0.680713 | 0.274067 | 0.221713 | 5.538441 | 0.618653 | 0.707664 | 61.714618 |
| | PASD(s20) | 24.608645 | 0.712806 | 0.242913 | 0.209351 | 4.795048 | 0.540107 | 0.625039 | 52.826575 |
| | Resshift(s15) | 24.932412 | 0.734073 | 0.198175 | 0.187559 | 5.319726 | 0.557401 | 0.677602 | 46.756755 |
| | Resshift(s4) | 25.262886 | 0.729898 | 0.207532 | 0.193287 | 5.930618 | 0.481287 | 0.646861 | 47.521572 |
| | SinSR(s1) | 24.814703 | 0.732119 | 0.264237 | 0.204537 | 5.189266 | 0.504561 | 0.639777 | 49.655490 |
| | Osediff(s1) | 23.221523 | 0.658442 | 0.279372 | 0.221037 | 5.464938 | 0.477634 | 0.661837 | 69.084736 |
| | InvSR(s1) | 23.163527 | 0.660025 | 0.274930 | 0.226382 | 4.827873 | 0.588372 | 0.704826 | 45.806334 |
| | Ours(s4) | 24.537216 | 0.730573 | 0.201882 | 0.198432 | 5.366422 | 0.649319 | 0.757744 | 44.208402 |
| DIV2k | HR | inf | 1.00000 | 0.00000 | 0.00000 | 4.87995 | 0.308124 | 0.458772 | 0.00000 |
| | Bicubic | 25.67952 | 0.71131 | 0.34384 | 0.320166 | 7.23905 | 0.190022 | 0.31281 | 91.37396 |
| | SwinIR | 25.524262 | 0.756364 | 0.196358 | 0.253459 | 6.574358 | 0.236129 | 0.369175 | 66.022553 |
| | Realesrgan | 25.017905 | 0.729274 | 0.208930 | 0.231028 | 4.377671 | 0.323017 | 0.440141 | 64.281207 |
| | StableSR(s200) | 25.089883 | 0.723934 | 0.230532 | 0.241219 | 6.143113 | 0.334181 | 0.402819 | 63.208495 |
| | DiffBIR(s50) | 24.952325 | 0.664533 | 0.245898 | 0.222816 | 5.170970 | 0.519603 | 0.694852 | 64.320329 |
| | PASD(s20) | 25.268907 | 0.730030 | 0.239548 | 0.222048 | 5.840237 | 0.409314 | 0.503350 | 66.624089 |
| | Resshift(s15) | 25.195770 | 0.682318 | 0.236521 | 0.230849 | 6.443043 | 0.367619 | 0.565044 | 69.377387 |
| | Resshift(s4) | 25.195317 | 0.682465 | 0.210933 | 0.236403 | 6.325634 | 0.324337 | 0.526237 | 68.193004 |
| | SinSR(s1) | 25.323979 | 0.700209 | 0.226813 | 0.238258 | 6.067024 | 0.359430 | 0.557111 | 87.956015 |
| | Osediff(s1) | 24.794832 | 0.714938 | 0.2194821 | 0.213805 | 5.859347 | 0.427836 | 0.689721 | 67.801938 |
| | InvSR(s1) | 24.143282 | 0.670584 | 0.2415381 | 0.253827 | 5.981039 | 0.471184 | 0.691187 | 74.120972 |
| | Ours(s4) | 24.910596 | 0.682398 | 0.215568 | 0.235888 | 6.024669 | 0.475017 | 0.707421 | 68.141436 |

Table 3: Quantitative evaluation of the fidelity–perceptual tradeoff obtained by sign-flip control of our correction scale parameter. For each dataset, we report results for HR-guided and LR-guided calibration with both positive and negative correction directions. Ours(s4, HR,-) is included only as a comparison case. These results confirm that our plug-in module enables continuous control over SR characteristics.

| Datasets | Methods | PSNR↑ | SSIM↑ | LPIPS↓ | DISTS↓ | NIQE↓ | MANIQA↑ | CLIPIQA↑ | FID↓ |
|---|---|---|---|---|---|---|---|---|---|
| RealSR | HR | inf | 1.00000 | 0.00000 | 0.00000 | 4.46300 | 0.48905 | 0.63147 | 0.00005 |
| | Bicubic | 25.93227 | 0.757129 | 0.28747 | 0.238012 | 6.76492 | 0.40826 | 0.50845 | 68.46397 |
| | Resshift(s4) | 25.262886 | 0.729898 | 0.207532 | 0.193287 | 5.930618 | 0.481287 | 0.646861 | 47.521572 |
| | Ours(s4, HR, +) | 27.563256 | 0.859821 | 0.169382 | 0.147571 | 4.736475 | 0.553831 | 0.651693 | 35.052837 |
| | Ours(s4, HR, -) | 13.272173 | 0.203726 | 0.595241 | 0.449212 | 20.379281 | 0.659814 | 0.687212 | 156.012784 |
| | Ours(s4, LR, +) | 25.692837 | 0.784613 | 0.172162 | 0.187129 | 5.211721 | 0.413489 | 0.503721 | 56.837271 |
| | Ours(s4, LR,-) | 24.537216 | 0.730573 | 0.201882 | 0.198432 | 5.366422 | 0.649319 | 0.757744 | 44.208402 |
| DIV2k | HR | inf | 1.00000 | 0.00000 | 0.00000 | 4.87995 | 0.308124 | 0.458772 | 0.00000 |
| | Bicubic | 25.67952 | 0.71131 | 0.34384 | 0.320166 | 7.23905 | 0.190022 | 0.31281 | 91.37396 |
| | Resshift(s4) | 25.195317 | 0.682465 | 0.210933 | 0.236403 | 6.325634 | 0.324337 | 0.526237 | 68.193004 |
| | Ours(s4, HR, +) | 27.69840 | 0.77613 | 0.15460 | 0.19407 | 5.75048 | 0.301922 | 0.45173 | 64.98641 |
| | Ours(s4, HR, -) | 12.87786 | 0.21311 | 0.51891 | 0.36488 | 10.45456 | 0.435000 | 0.62238 | 125.65250 |
| | Ours(s4, LR, +) | 25.438372 | 0.691362 | 0.209572 | 0.221728 | 5.821021 | 0.251827 | 0.408611 | 65.911252 |
| | Ours(s4,LR,-) | 24.910596 | 0.682398 | 0.215568 | 0.235888 | 6.024669 | 0.475017 | 0.707421 | 68.141436 |

Table 4: (A) Absolute Error changes under different scale-parameter configurations on Div2K dataset in the 4-step model. Row [3] corresponds to the optimally calibrated scale parameters that minimize the loss function and achieve error correction. Negative configurations in Row [1] and [2] increase the absolute error as expected, while row [4] shows that excessively large parameters may also fail to reduce error. (B) optimal parameter configuration for step 15 models.

| [] | scale parameters | Step 3 | Step 2 | Step 1 |
|---|---|---|---|---|
| [1] | $(-2500, -120, -100)$ | -160 | -1679 | -8428 |
| [2] | $(-1700, -70, -60)$ | -71 | -750 | -3374 |
| [3] | $(1700, 70, 60)$ | 78 | 607 | 3036 |
| [4] | $(2500, 120, 100)$ | 89 | -115 | 879 |

Table 5: Computation costs of existing methods to process 1000 tiles(64x64) with from DIV2K dataset.

|  | SwinIR | Esrgan | Realesrgan | StableSR | DiffBir | SeeSR | PASD | ResShift(s15) | ResShift(s4) | OseDiff | InvSR | Ours(s4) |
|---|---|---|---|---|---|---|---|---|---|---|---|---|
| Time(s) | 191 | 23 | 12 | 148 | 1948 | 994 | 3105 | 808 | 229 | 181 | 115 | 235 |

# 6 NUMERICAL ERROR RESTORATION: REFERENCE-BASED AND REFERENCE-FREE APPROACHES

## 6.1 RESTORATION IN REFERENCE REGION AND NON-REFERENCE REGION

We first estimate correction parameters $A_t$ and $B_t$ in reference regions by linear regression of numerical errors and apply them via classifier guidance. As shown in Table 3, fidelity enhancement (e.g., Ours(hr,+)) requires guidance forms such as option 1 or 5 in Fig. 4. However, in non-reference regions, only options 3 or 7 (LR-only regression) can be used, which yield overly smoothed results. Fig. 5 shows that cosine similarity between LR–SR and SR–HR trajectories remains below 0.8 in most regions, confirming misalignment. Thus, fidelity-oriented correction is infeasible in non-reference regions. Exploring pretrained models with better trajectory alignment is left as a future work.

**Fidelity vs. Perception Trade-off** Numerical correction improves reference-based metrics (PSNR, SSIM, LPIPS) but not perceptual quality. To address this, we flip the sign of the correction term. As seen in Fig. 4, cases 2,4,6,8 do not increase fidelity but yield sharper perceptual appearance, even for LR-only guidance (cases 6 or 8). In contrast, direct fidelity correction (cases 1 or 5) recovers small details (e.g., text) but increases blur. We therefore propose to use sign-flipped image-based classifier guidance as a practical plug-in mechanism for perceptual enhancement in non-reference regions.

**Calibration Strategy** Figure 6 illustrates the structural complexity involved in reliably calibrating this parameter. As described in Section 4, the calibration process relies on accurate estimation of the model's discretization error. For Table 2, parameters are calibrated using 500 images. However, Figure 6 highlights an important characteristic of this procedure: when parameters are averaged over a large collection of images, the resulting calibration value may diverge substantially from the image-specific optimal values.

Each curve in Figure 6 represents the error trajectory obtained from a single LR–HR tile pair. The index at which each curve achieves its minimum exhibits significant variation across tiles, reflecting the diversity of local image statistics. For both RealSR and DIV2K, the magenta line denotes the index corresponding to the minimum of the accumulated error curves, while the gray line represents the averaged calibration used in Table 2. Although the average indices of DIV2K and RealSR differ from the ImageNet-based calibration, the discrepancy is smaller for DIV2K, explaining the comparatively reduced quality variation observed for DIV2K in Table 3.

This deviation between averaged and image-specific optima has direct consequences. When a tile's true minimum lies below the average index, applying the average parameter produces limited perceptual enhancement. Conversely, when it lies above, the correction becomes aggressive, sometimes resulting in pronounced artifacts (see Figure 13, examples 4-5th columns). Using the global minimum avoids these effects but suppresses enhancement for most images, indicating that *per-image calibration is the most reliable approach*.

While per-image calibration is generally infeasible in blind SR, it becomes practical when reference views are available (e.g., multi-camera systems Lee et al. (2022)). Table 6 shows that tile-inferred per-image parameters reduce variance by approximately 50% relative to dataset-level averaging, being able to significantly improve visual consistency.

The parameters reported in Figure 13 differ substantially across images, and even moderate deviations $\approx 50\%$ from optimal values significantly influence visual characteristics. For the 4-step model, three scale parameters define the control space because the predefined variance at step 0 is zero; the 15-step model requires fourteen parameters, further emphasizing the importance of structured calibration.

## 6.2 PERFORMANCE COMPARISON

Table 5 reaffirms the value of our approach. Although ResShift s4 version has more numerical errors than s15 version and makes it less favorable in image quality metrics, it has the advantages in the limited runtime environment. However, after applying our method, the image quality metrics become comparable to those of s15 version, while the speed remains at the s4 level. Therefore, our approach serves as an option for maintaing optimal image quality while satisfying limited runtime constraints.

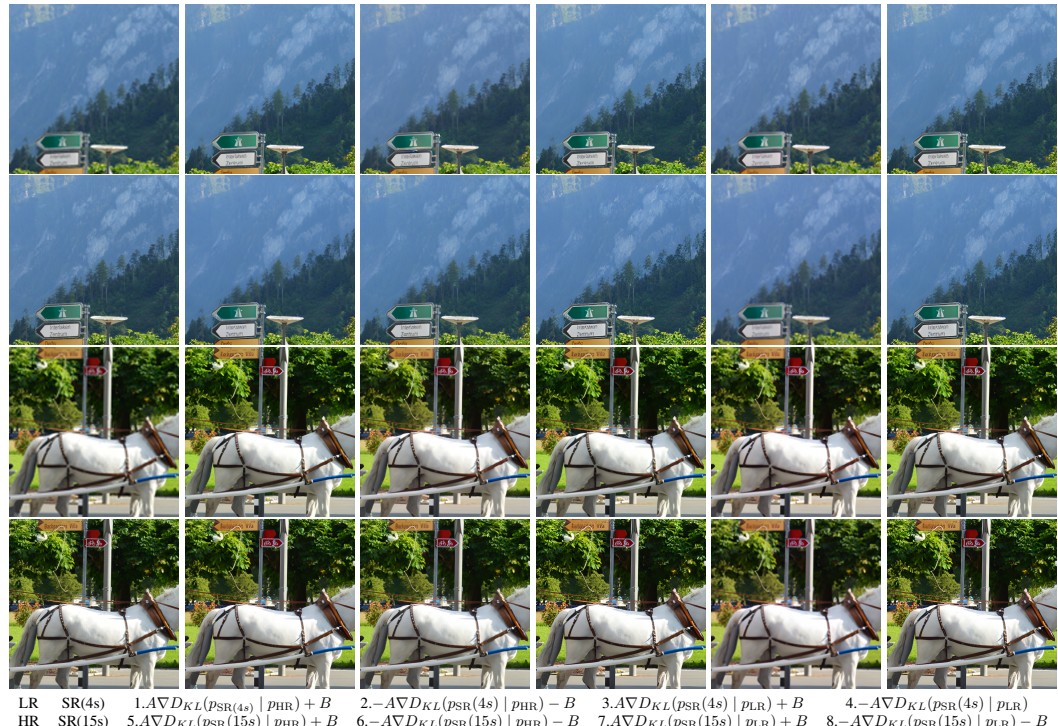

LR  SR(4s)  $1. A\nabla D_{KL}(p_{\text{SR}(4s)} \mid p_{\text{HR}}) + B$   $2. -A\nabla D_{KL}(p_{\text{SR}(4s)} \mid p_{\text{HR}}) - B$   $3. A\nabla D_{KL}(p_{\text{SR}(4s)} \mid p_{\text{LR}}) + B$   $4. -A\nabla D_{KL}(p_{\text{SR}(4s)} \mid p_{\text{LR}})$

HR  SR(15s)  $5. A\nabla D_{KL}(p_{\text{SR}(15s)} \mid p_{\text{HR}}) + B$   $6. -A\nabla D_{KL}(p_{\text{SR}(15s)} \mid p_{\text{HR}}) - B$   $7. A\nabla D_{KL}(p_{\text{SR}(15s)} \mid p_{\text{LR}}) + B$   $8. -A\nabla D_{KL}(p_{\text{SR}(15s)} \mid p_{\text{LR}}) - B$

Figure 4: Reference region restoration result by various guidance schemes. The guidance schemes applied to the images above are summarized in the table below. Detailed description is available in section 6.1

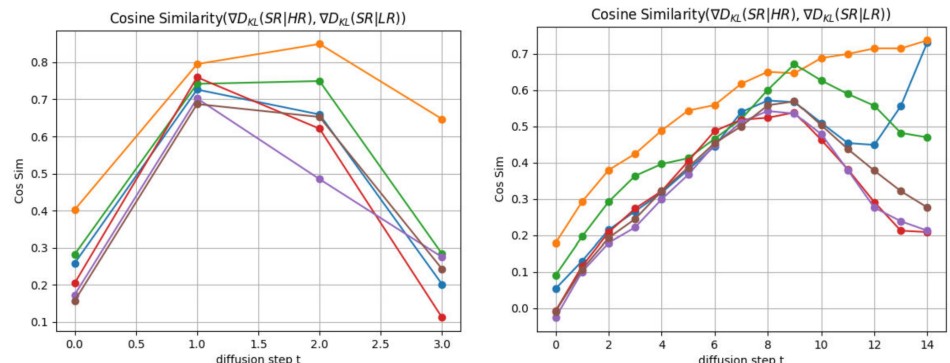

Figure 5: (a) cosine similarity between the trajectory between $\nabla D_{KL}(p_{SR}|p_{HR})$ and $\nabla D_{KL}(p_{SR}|p_{LR})$ for step 4 and (b) for step 15 pretrained models of Yue et al. (2023)

## 6.3 ABLATION STUDY

In Figure 9, we present the error plots for the s4 and s15 version. The overall error reduction is highly sensitive to the choice of optimized parameters: negative values of the parameters tend to increase cumulative error, while appropriately optimized parameters significantly reduce it.

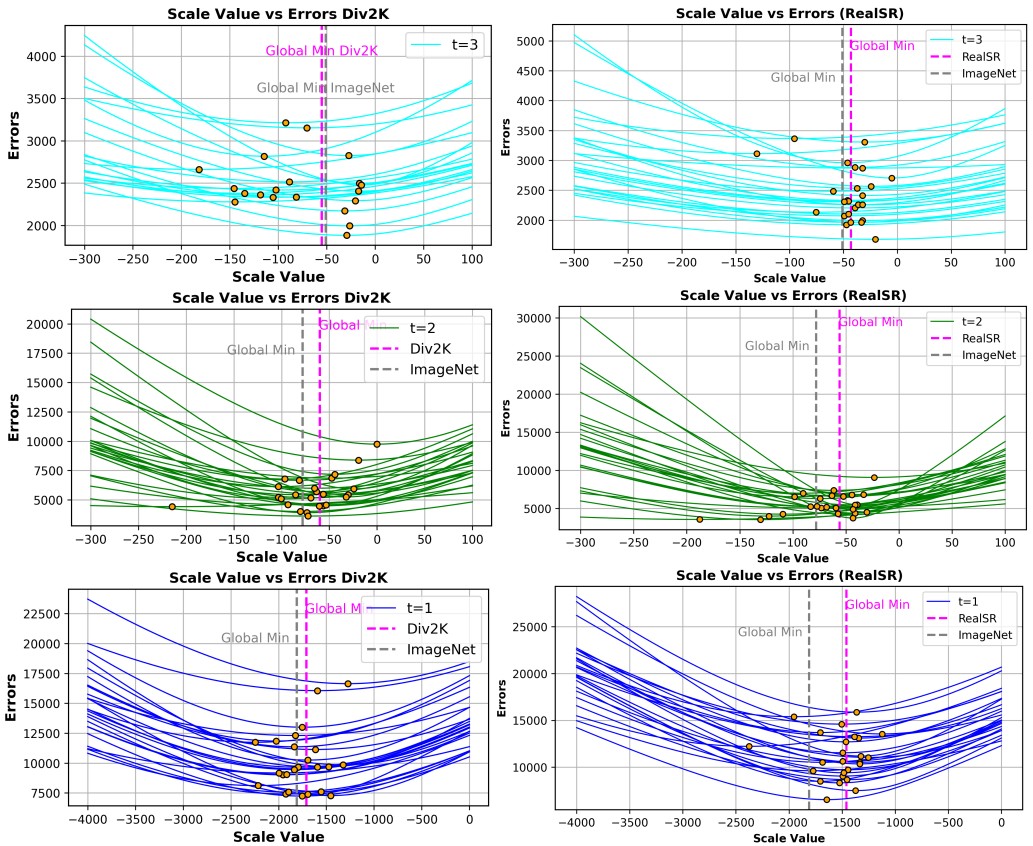

Figure 6: optimal parameters are negative values which corresponds to A in figure 4. we apply $-A\nabla D_{KL}(p_{\mathrm{SR}} \mid p_{\mathrm{LR}})$ and so positive value will be applied in the end. at $t = 0$, variance is zero and so guidance becomes zero as well.

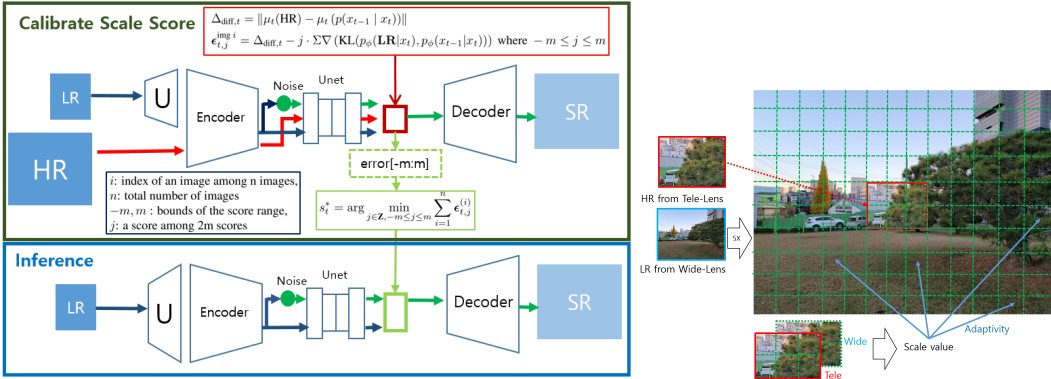

Figure 7: (a) Block diagram for inferring the non-overlapped region using optimized parameters extracted from the overlapped region. A score vector $\mathbf{S}^*$ means the scale score calibrated in Algorithm 1 stage. Calibrated score vector is delivered to Inference pipeline described in Algorithm 2. (b) This figure illustrates a plug-in module pipeline that applies tele camera-calibrated parameters to all tile region.

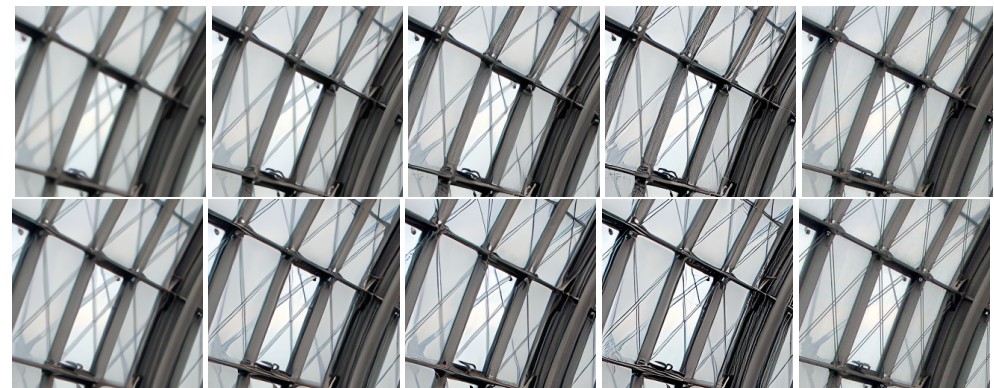

Figure 8: Non-overlap region. Image quality comparison for various scale value. The images from the left to the right have scale values such as $(\mathbf{A}^*)$, $(-0.5 \cdot \mathbf{A}^*)$, $(-\mathbf{A}^*)$, $(-2 \cdot \mathbf{A}^*)$, HR. The first row uses step 4 and the second row uses step 15

## 7 MATHEMATICAL DETAILS

**Lemma 3.2.1 (KL gradient in the Gaussian case)**  For two normal distributions $\mathcal{P} : \mathcal{N}(\mathbf{x}_t, \sigma_1^2)$, $\mathcal{Q} : \mathcal{N}(\boldsymbol{\mu}_t, \sigma_2^2)$

$$\nabla_{\mathbf{x}} D_{\mathrm{KL}}\big(\mathcal{P}||\mathcal{Q})\big) \approx \frac{(\mathbf{x}_t - \boldsymbol{\mu}_t)}{\sigma_2^2}.$$

**Proof.**  From Soch (2024), Kullback-Leibler divergence for two normal distributions $\mathcal{P}$ and $\mathcal{Q}$ is as follows:

$$D_{\mathrm{KL}}[\mathcal{P}||\mathcal{Q}] = \frac{1}{2}\left(\log\frac{\sigma_2^2}{\sigma_1^2} + \frac{\sigma_1^2}{\sigma_2^2} + \frac{(\mathbf{x}_t - \boldsymbol{\mu}_2)^2}{2\sigma_2^2}\right)$$

We assume same variance in two distributions. For two distributions $D_{KL}(\mathcal{N}(\mathbf{x}, \sigma_1^2), \mathcal{N}(\boldsymbol{\mu}_t, \sigma_2^2))$, we get their KL divergence as follows.

$$\nabla_{\mathbf{x}}\mathrm{KL} = \nabla_{\mathbf{x}}\left[\mathrm{KL}[\mathcal{P}||\mathcal{Q}]\right] = \frac{\mathbf{x}_t - \boldsymbol{\mu}_t}{\sigma_2^2}.$$

**Definition 3.2.1 (Second-order derivative as the numerical error)**  We define the *numerical error* (DE) of the first-order derivative solution as

$$\mathbf{x}_t^{\mathrm{Exact}} - \mathbf{x}_t^{\mathrm{Euler(1st)}} \approx \nabla_t\big[f(t)\,\mathbf{x}_t - \tfrac{1}{2}g^2(t)\,\nabla_{\mathbf{x}}\log p_t(\mathbf{x}_t)\big].$$

**Proof**  From Eq. 1, discretizing with a first-order Euler step of size $\Delta t$:

$$\mathbf{x}_{t+\Delta t}^{(\mathrm{Euler})} = \mathbf{x}_t + \Delta t\left[f(t)\,\mathbf{x}_t - \tfrac{1}{2}g^2(t)\,\nabla_{\mathbf{x}}\log p_t(\mathbf{x}_t)\right].$$

Comparing to the exact solution

$$\mathbf{x}_{t+\Delta t}^{(\mathrm{Exact})} = \mathbf{x}_t + \int_t^{t+\Delta t}\left[f(s)\,\mathbf{x}_s - \tfrac{1}{2}g^2(s)\,\nabla_{\mathbf{x}}\log p_s(\mathbf{x}_s)\right]ds,$$

Taylor expansion of the above equation yields

$$\mathbf{x}(t) \approx \mathbf{x}_s + (t-s)V + \frac{1}{2}(t-s)^2\nabla_t V + \mathcal{O}((t-s)^3),$$
$$\text{where } V = f(t)\mathbf{x}_t - \tfrac{1}{2}g^2(t)\nabla_{\mathbf{x}}\log p_t(\mathbf{x}_t). \tag{12}$$

A second-order (i.e., $\mathcal{O}(\Delta t^2)$) discretization error term appears upon Taylor expansion.

$$\mathbf{x}_{t+\Delta t}^{(\mathrm{Exact})} = \mathbf{x}_t + \Delta t V + \frac{1}{2}(\Delta t)^2\nabla_t V + \mathcal{O}((\Delta t)^3)$$
$$\mathbf{x}_{t+\Delta t}^{(\mathrm{Euler})} = \mathbf{x}_t + \Delta t V + \mathcal{O}((\Delta t)^2) \tag{13}$$

Subtraction of two above equations yields

$$\mathrm{DE} = \mathbf{x}_{t+\Delta t}^{(\mathrm{Exact})} - \mathbf{x}_{t+\Delta t}^{(\mathrm{Euler})} \approx \tfrac{1}{2}(\Delta t)^2\nabla_t V \approx \nabla_t\big[f(t)\mathbf{x}_t - \tfrac{1}{2}g^2(t)\nabla_{\mathbf{x}}\log p_t(\mathbf{x}_t)\big].$$

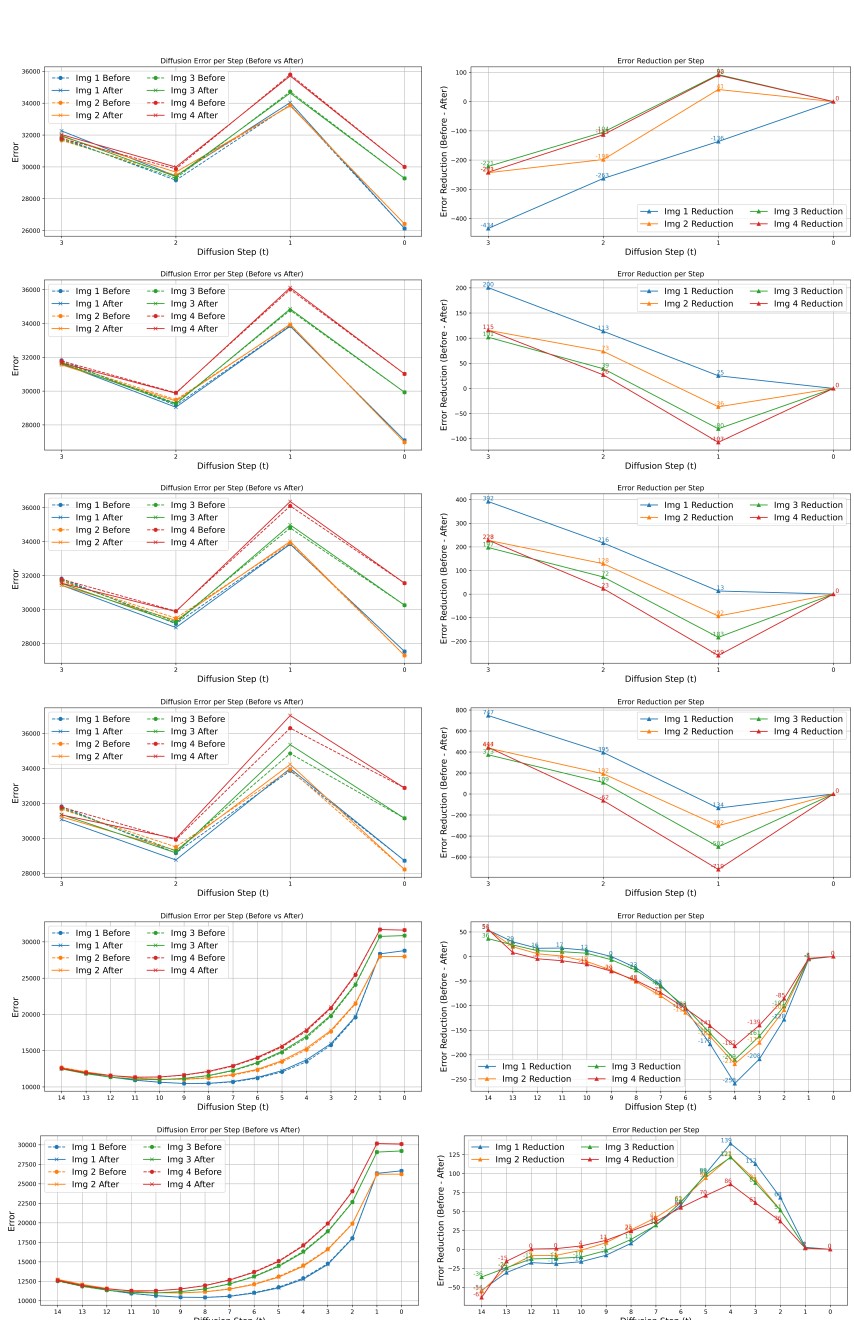

Figure 9: Non-overlap region. Cumulative Error at each diffusion step. Left Column denotes $\mathbf{E}_t^{\mathrm{cumu}}$ and Right Column denotes $\mathbf{E}_t^{\mathrm{cumu}}$(before)-$\mathbf{E}_t^{\mathrm{cumu}}$(after). According to the row sequence: $(1) -A^*$ $(2)$ $0.5 \cdot A^*$ $(3)$ $1 \cdot A^*$ $(4)$ $2 \cdot A^*$ $(5) -A^*$ $(6)$ $A^*$

**Proposition 3.2.1 (Constant Variance Leads to a Simpler Form).** Let $\sigma_t^2$ be a predefined constant over the spatial dimensions. Then, the gradient with respect to $t$ of the modified score function from Definition 3.2.1 satisfies

$$\nabla_t \left[ f(t)\,\mathbf{x}_t - \tfrac{1}{2}g^2(t)\,\nabla_{\mathbf{x}} \log p_t(\mathbf{x}_t) \right] \approx A_t \cdot \frac{(\mathbf{x}_t - \boldsymbol{\mu}_t)}{\sigma_t^2} + B_t.$$

where $A_t, B_t \in \mathbb{R}$ are step-dependent coefficients to minimize discretization error.

**Proof** We apply the total derivative:

$$\frac{d}{dt}\left[ f(t)\mathbf{x}(t) - \tfrac{1}{2}g^2(t)\nabla_{\mathbf{x}} \log p_t(\mathbf{x}) \right] = \frac{d}{dt}\left[ f(t)\mathbf{x}(t) \right] - \frac{d}{dt}\left[ \tfrac{1}{2}g^2(t)\nabla_{\mathbf{x}} \log p_t(\mathbf{x}) \right].$$

First, since $\mathbf{x} = \mathbf{x}(t)$, the chain rule gives:

$$\frac{d}{dt}[f(t)\mathbf{x}(t)] = f'(t)\mathbf{x} + f(t)\frac{d\mathbf{x}}{dt} = f'(t)\mathbf{x} + f(t)\left( f(t)\mathbf{x} - \tfrac{1}{2}g^2(t)\nabla_{\mathbf{x}} \log p_t(\mathbf{x}) \right).$$

Now compute the second term:

$$\frac{d}{dt}[g^2(t)\nabla_{\mathbf{x}} \log p_t(\mathbf{x})] = 2g(t)g'(t)\nabla_{\mathbf{x}} \log p_t(\mathbf{x}) + g^2(t)\frac{\partial}{\partial t}\left( \nabla_{\mathbf{x}} \log p_t(\mathbf{x}) \right).$$

We now assume $p_t(\mathbf{x}) = \mathcal{N}(\boldsymbol{\mu}_t, \sigma_t^2 I)$, so that

$$\nabla_{\mathbf{x}} \log p_t(\mathbf{x}) = -\frac{1}{\sigma_t^2}(\mathbf{x} - \boldsymbol{\mu}_t),$$

and hence:

$$\frac{\partial}{\partial t}\nabla_{\mathbf{x}} \log p_t(\mathbf{x}) = -\left[ \frac{d}{dt}\left( \frac{1}{\sigma_t^2} \right)(\mathbf{x} - \boldsymbol{\mu}_t) - \frac{1}{\sigma_t^2}\boldsymbol{\mu}_t' \right].$$

Now substitute all terms:

$$\nabla_t \left[ f(t)\mathbf{x} - g^2(t)\nabla_{\mathbf{x}} \log p_t(\mathbf{x}) \right] = f'(t)\mathbf{x} + f^2(t)\mathbf{x} - f(t)g^2(t)\nabla_{\mathbf{x}} \log p_t(\mathbf{x})$$
$$- g(t)g'(t)\nabla_{\mathbf{x}} \log p_t(\mathbf{x}) - \tfrac{1}{2}g^2(t)\left[ -\frac{d}{dt}\left( \frac{1}{\sigma_t^2} \right)(\mathbf{x} - \boldsymbol{\mu}_t) + \frac{1}{\sigma_t^2}\boldsymbol{\mu}_t' \right]$$

Substitute the gradient expression:

$$\nabla_{\mathbf{x}} \log p_t(\mathbf{x}) = -\frac{1}{\sigma_t^2}(\mathbf{x} - \boldsymbol{\mu}_t),$$

we collect terms as:

$$\nabla_t \left[ f(t)\mathbf{x} - g^2(t)\nabla_{\mathbf{x}} \log p_t(\mathbf{x}) \right] = (f'(t) + f^2(t))\mathbf{x} + \left[ \frac{f(t)g^2(t) + 2g(t)g'(t)}{\sigma_t^2} \right](\mathbf{x} - \boldsymbol{\mu}_t)$$
$$+ \left[ g^2(t)\frac{d}{dt}\left( \frac{1}{\sigma_t^2} \right)(\mathbf{x} - \boldsymbol{\mu}_t) \right] + \frac{g^2(t)}{\sigma_t^2}\boldsymbol{\mu}_t'$$
$$= \mathsf{A}(t) \cdot \mathbf{x} + \mathsf{B}(t) \cdot (\mathbf{x} - \boldsymbol{\mu}_t) + g^2(t)\frac{d}{dt}\left( \frac{\mathbf{x}}{\sigma_t^2} \right) + \mathsf{C}(t)$$

where $\mathsf{A}, \mathsf{B}, \mathsf{C}$ denote the aggregated terms consisting of corresponding constant components. By substituting $\frac{d}{dt}\left( \frac{\mathbf{x}}{\sigma_t^2} \right)$ with the following equation,

$$\frac{d}{dt}\left( \frac{\mathbf{x}}{\sigma_t^2} \right) = \frac{f(t)}{\sigma_t^2}\mathbf{x} + \frac{\mathbf{x} - \boldsymbol{\mu}_t}{\sigma_t^4} - \frac{2 \cdot \mathbf{x}}{\sigma_t^2} = \mathsf{D}(t)(\mathbf{x} - \boldsymbol{\mu}_t) + \mathsf{E}(t)$$

Now, we have the decomposition:

$$\nabla_t \left[ f(t)\mathbf{x} - g^2(t)\nabla_{\mathbf{x}} \log p_t(\mathbf{x}) \right] = \alpha(t)(\mathbf{x} - \boldsymbol{\mu}_t) + \beta(t) \approx A_t \cdot (\mathbf{x} - \boldsymbol{\mu}_t) + B_t$$

**Proposition 3.2.2 (Approximation Error and KL Gradient Scaling).** For a given step $t$, the difference between the exact solution $\mathbf{x}^{\text{Exact}}$ and the Euler discretization $\mathbf{x}^{\text{Euler}}$ can be expressed as

$$\mathbf{x}_t^{\text{Exact}} - \mathbf{x}_t^{\text{Euler}} \approx A_t \cdot \nabla_{\mathbf{x}} D_{\text{KL}}(\mathcal{P} \,\|\, \mathcal{Q}) + B_t.$$

where $\mathcal{P}$ is discrete distribution, $\mathcal{Q}$ is continuous exact distribution, $\mathcal{P} : \mathcal{N}(\mathbf{x}_t, \sigma_t^2), \mathcal{Q} : \mathcal{N}(\boldsymbol{\mu}_t, \sigma_t^2)$.

**Proof.** From Definition 3.2.1, the discretization error satisfies:

$$\mathbf{x}_t^{\text{Exact}} - \mathbf{x}_t^{\text{Euler}} \approx \nabla_t \big[ f(t)\mathbf{x}_t - \tfrac{1}{2} g^2(t) \nabla_{\mathbf{x}} \log p_t(\mathbf{x}_t) \big].$$

Applying Proposition 3.2.1 under the Gaussian assumption:

$$\mathbf{x}_t^{\text{Exact}} - \mathbf{x}_t^{\text{Euler}} \approx A_t \cdot \frac{(\mathbf{x}_t - \boldsymbol{\mu}_t)}{\sigma_t^2} + B_t.$$

Meanwhile, from Lemma 3.2.1 with $P = N(x_t, \sigma_t^2)$ and $Q = N(\mu_t, \sigma_t^2)$:

$$\nabla_{\mathbf{x}} D_{\text{KL}}(\mathcal{P} \,\|\, \mathcal{Q}) = \frac{(\mathbf{x}_t - \boldsymbol{\mu}_t)}{\sigma_t^2}.$$

Substituting this into the error expression yields:

$$\mathbf{x}_t^{\text{Exact}} - \mathbf{x}_t^{\text{Euler}} \approx A_t \cdot \nabla_{\mathbf{x}} D_{\text{KL}}(\mathcal{P} \,\|\, \mathcal{Q}) + B_t. \quad \square$$

**Proposition 3.2.3 (Alternative KL Usage Under Lipschitz Continuity).** Under a suitable Lipschitz continuity assumption, the KL divergence term $D_{\text{KL}}(p_t \,\|\, q_{\text{HR}})$ may be replaced with $D_{\text{KL}}(p_t \,\|\, q_{\text{LR}})$, and moreover,

$$\nabla_{\mathbf{x}} D_{\text{KL}}(p_t \,\|\, q_{\text{HR}}) \approx \nabla_{\mathbf{x}} D_{\text{KL}}(p_t \,\|\, q_{\text{LR}}).$$

**Proof.** Let $q_{\text{HR}} = q(\mathbf{x}_t | \mathbf{x}_{\text{HR}})$ and $q_{\text{LR}} = q(\mathbf{x}_t | \mathbf{x}_{\text{LR}})$ denote the forward noising kernels. For convenience, denote $p_t = p(\mathbf{x}_t)$. The forward diffusion process is Gaussian:

$$q(\mathbf{x}_t | \mathbf{x}_0) = \mathcal{N}\big( \sqrt{\bar{\alpha}_t} \mathbf{x}_0, \, (1 - \bar{\alpha}_t) I \big).$$

Thus,

$$q(\mathbf{x}_t | \mathbf{x}_{\text{HR}}) = \mathcal{N}\big( \sqrt{\bar{\alpha}_t} \mathbf{x}_{\text{HR}}, \, (1 - \bar{\alpha}_t) I \big),$$
$$q(\mathbf{x}_t | \mathbf{x}_{\text{LR}}) = \mathcal{N}\big( \sqrt{\bar{\alpha}_t} \mathbf{x}_{\text{LR}}, \, (1 - \bar{\alpha}_t) I \big).$$

Because the covariances are identical, the closed-form Wasserstein–2 distance Gelbrich (1990) reduces to

$$\mathbf{W}_2\big( q_{\text{HR}}, q_{\text{LR}} \big) = \sqrt{\bar{\alpha}_t} \, \| \mathbf{x}_{\text{HR}} - \mathbf{x}_{\text{LR}} \|_2 =: \delta_t.$$

On the class of Gaussian distributions with bounded covariance, the KL functional is locally Lipschitz with respect to the $\mathbf{W}_2$ metric Ambrosio et al. (2005); Villani (2008). Hence, there exists $C > 0$ such that

$$\big| D_{\text{KL}}(p_t \| q_{\text{HR}}) - D_{\text{KL}}(p_t \| q_{\text{LR}}) \big| \leq C \, \mathbf{W}_2(q_{\text{HR}}, q_{\text{LR}}) = C \, \delta_t.$$

Since KL$(p_t \| q)$ is smooth in the mean parameter of $q$, its spatial gradient is also Lipschitz-continuous: there exists $L > 0$ such that

$$\big\| \nabla_{\mathbf{x}} D_{\text{KL}}(p_t \| q_{\text{HR}}) - \nabla_{\mathbf{x}} D_{\text{KL}}(p_t \| q_{\text{LR}}) \big\| \leq L \, \mathbf{W}_2(q_{\text{HR}}, q_{\text{LR}}) = L \, \delta_t.$$

Because $q_{\text{HR}}$ and $q_{\text{LR}}$ differ only in the mean parameter, $\delta_t$ depends solely on $\| \mathbf{x}_{\text{HR}} - \mathbf{x}_{\text{LR}} \|_2$. Thus,

$$\nabla_{\mathbf{x}} D_{\text{KL}}(p_t \| q_{\text{HR}}) \approx \nabla_{\mathbf{x}} D_{\text{KL}}(p_t \| q_{\text{LR}}),$$

whenever $\| \mathbf{x}_{\text{HR}} - \mathbf{x}_{\text{LR}} \|_2$ is small.

**Proposition 3.3.1 (Guidance for LR-based Conditioning)** The target distribution with LR guidance is given by

$$\mathcal{N}\Big( \boldsymbol{\mu} \, + \, \Sigma \, \nabla p_\theta(\mathbf{c} \mid \mathbf{x}_t, \mathbf{y}_0, t), \, \Sigma \Big), \tag{14}$$

where $\mathbf{c}$ indicates the LR guidance.

**Proof** First, we enumerates derivation process of ResShift algorithm Yue et al. (2023) here.

$$q(\mathbf{x}_t|\mathbf{x}_{t-1}, \mathbf{y}_0) = \mathcal{N}(\mathbf{x}_t; \mathbf{x}_{t-1} + \alpha_t e_0, \kappa^2 \alpha_t I), \tag{15}$$

where $e_0 = \mathbf{y}_0 - \mathbf{x}_0$ is the residual between the c and HR images, and $\kappa$ is a hyper-parameter controlling the noise variance. The reverse process aims to estimate the posterior distribution $p(\mathbf{x}_0|\mathbf{y}_0)$ by follows:

$$p(\mathbf{x}_0|\mathbf{y}_0) = \int p(\mathbf{x}_T|\mathbf{y}_0) \prod_{t=1}^{T} p_\theta(\mathbf{x}_{t-1}|\mathbf{x}_t, \mathbf{y}_0) \mathrm{d}\mathbf{x}_{1:T}, \tag{16}$$

The targeted distribution is as follows:

$$p_\theta(\mathbf{x}_{t-1}|\mathbf{x}_t, \mathbf{y}_0) = \mathcal{N}\left(\mathbf{x}_{t-1}; \mu_\theta(\mathbf{x}_t, \mathbf{y}_0, t), \Sigma(t)\right) \tag{17}$$

$$\Sigma(t) = \kappa^2 \frac{\eta_{t-1}}{\eta_t} \alpha_t I. \tag{18}$$

$$\mu_\theta(\mathbf{x}_t, \mathbf{y}_0, t) = \frac{\eta_{t-1}}{\eta_t} \mathbf{x}_t + \frac{\alpha_t}{\eta_t} f_\theta(\mathbf{x}_t, \mathbf{y}_0, t), \tag{19}$$

where $\alpha_t = \eta_t - \eta_{t-1}$ for $t > 1$ and $\alpha_1 = \eta_1$.

In the previous section, we proposed conditional diffusion process as follows:

$$p(\mathbf{x}_0|\mathbf{y}_0, \mathbf{c}) = \int p(\mathbf{x}_T|\mathbf{y}_0, \mathbf{c}) \prod_{t=1}^{T} p_\theta(\mathbf{x}_{t-1}|\mathbf{x}_t, \mathbf{y}_0, \mathbf{c}) \mathrm{d}\mathbf{x}_{1:T}, \tag{20}$$

In the above equation, $p_\theta(\mathbf{x}_{t-1}|\mathbf{x}_t, \mathbf{y}_0, \mathbf{c})$ can be simplified.

$$
\begin{aligned}
p_\theta(\mathbf{x}_{t-1}|\mathbf{x}_t, \mathbf{y}_0, c) &= \frac{p(\mathbf{x}_{t-1}, \mathbf{x}_t, \mathbf{y}_0, c)}{p(\mathbf{x}_t, \mathbf{y}_0, c)} &&= \frac{p(\mathbf{x}_{t-1}, \mathbf{x}_t, \mathbf{y}_0, c)}{p(\mathbf{c}|\mathbf{x}_t, \mathbf{y}_0)p(\mathbf{x}_t, \mathbf{y}_0)} \\
&= \frac{p(\mathbf{c}|\mathbf{x}_{t-1}, \mathbf{x}_t, \mathbf{y}_0)p(\mathbf{x}_{t-1}, \mathbf{x}_t, \mathbf{y}_0)}{p(\mathbf{c}|\mathbf{x}_t, \mathbf{y}_0)p(\mathbf{x}_t, \mathbf{y}_0)} &&= \frac{p(\mathbf{c}|\mathbf{x}_{t-1}, \mathbf{x}_t, \mathbf{y}_0)p(\mathbf{x}_{t-1}|\mathbf{x}_t, \mathbf{y}_0)p(\mathbf{x}_t, \mathbf{y}_0)}{p(\mathbf{c}|\mathbf{x}_t, y)p(\mathbf{x}_t, \mathbf{y}_0)} \\
&= \frac{p(\mathbf{c}|\mathbf{x}_{t-1}, \mathbf{x}_t, \mathbf{y}_0)p(\mathbf{x}_{t-1}|\mathbf{x}_t, \mathbf{y}_0)}{p(\mathbf{c}|\mathbf{x}_t, \mathbf{y}_0)} &&= \frac{p(\mathbf{c}|\mathbf{x}_{t-1}, \mathbf{y}_0)p(\mathbf{x}_{t-1}|\mathbf{x}_t, \mathbf{y}_0)}{p(\mathbf{c}|\mathbf{x}_t)}
\end{aligned} \tag{21}
$$

The $p(\mathbf{c}|\mathbf{x}_t)$ term can be treated as a constant since it does not depend on $\mathbf{x}_{t-1}$. from Markov Chain property, $p(\mathbf{y}_0|\mathbf{x}_t, \mathbf{x}_{t+1}) = p(\mathbf{y}_0|\mathbf{x}_t)$ and $p(\mathbf{c}|\mathbf{x}_{t-1}, \mathbf{x}_t, \mathbf{y}_0) = p(\mathbf{c}|\mathbf{x}_{t-1}, \mathbf{y}_0)$ is derived. Now, we conclude as follows:

$$p_\theta(\mathbf{x}_{t-1}|\mathbf{x}_t, \mathbf{y}_0, \mathbf{c}) = Z p(\mathbf{x}_{t-1}|\mathbf{x}_t, \mathbf{y}_0) p(\mathbf{c}|\mathbf{x}_{t-1}, \mathbf{y}_0) \tag{22}$$

we already knows $p(\mathbf{x}_{t-1}|\mathbf{x}_t, \mathbf{y}_0)$ and we need to handle $p(\mathbf{c}|\mathbf{x}_{t-1}, \mathbf{y}_0)$. For now, we write it as $p(\mathbf{c}|\mathbf{x}_t, \mathbf{y}_0)$ in convenience. $\log p(\mathbf{c}|\mathbf{x}_t, \mathbf{y}_0)$ can be approximated by using a Taylor expansion around $\mathbf{x}_t = \mu$.

$$\log p(\mathbf{c}|\mathbf{x}_t, \mathbf{y}_0) \approx \log p(\mathbf{c}|\mathbf{x}_t, \mathbf{y}_0)|_{\mathbf{x}_t=\mu} + (\mathbf{x}_t - \mu)\nabla_{\mathbf{x}_t} \log p(\mathbf{c}|\mathbf{x}, \mathbf{y}_0)|_{\mathbf{x}_t=\mu}$$

Let $G = \nabla_{\mathbf{x}_t} \log p(\mathbf{c}|\mathbf{x}_t, \mathbf{y}_0)|_{\mathbf{x}_t=\mu}$. From Eq. 18 Eq. 19, we aleady got $\Sigma_\theta$ and $\mu$.

$$
\begin{aligned}
\log p_\theta(\mathbf{x}_{t-1}|\mathbf{x}_t, \mathbf{y}_0, \mathbf{c}) &\approx -\frac{1}{2}(\mathbf{x}_t - \mu)^T \Sigma^{-1}(\mathbf{x}_t - \mu) + (\mathbf{x}_t - \mu)G + C \\
&= -\frac{1}{2}(\mathbf{x}_t - \mu - \Sigma G)^T \Sigma^{-1}(\mathbf{x}_t - \mu - \Sigma G) + C
\end{aligned}
$$

Throughout all the above equations, the original $\mathbf{c}$ image can be fed by the condition $\mathcal{N}(\mu + \Sigma \nabla p_\theta(\mathbf{c}|\mathbf{x}_t, \mathbf{y}_0, t), \Sigma)$.

**Remark 3.3.1 (Implementation and Partial Guidance).** From Proposition 3.2.2 and Proposition 3.2.2, one obtains:

$$\nabla_{\mathbf{x}_t} \log p(\mathbf{c} \mid \mathbf{x}_t, \mathbf{y}_0) = \nabla D_{\mathrm{KL}}\left(p_\theta(\mathbf{x}_t) \,\|\, p_\theta(\mathbf{c})\right),$$

for an image-based guidance approach. Here we assume $\mathbf{x}_{\mathrm{LR}}$ is the vector at $\mathbf{x}_T$ prior to the diffusion process.

**Proof.** According to Lemma 3.2.1, let $\mathcal{P} = \mathcal{N}(\mathbf{x}, \sigma_1^2)$ and $\mathcal{Q} = \mathcal{N}(\boldsymbol{\mu}_t, \sigma_2^2)$. Then

$$\nabla D_{\mathrm{KL}}(\mathcal{P} \,\|\, \mathcal{Q}) \approx \frac{\mathbf{x} - \boldsymbol{\mu}_t}{\sigma^2}.$$

Suppose a final distribution is $\mathcal{N}(\mathbf{x} - \boldsymbol{\mu} - \Sigma G, \ \Sigma)$. We introduce two distributions $\mathcal{P}' = \mathcal{N}(\mathbf{x} - \boldsymbol{\mu}_t, \sigma^2)$ for $p(\mathbf{x}_0 \mid \mathbf{y}_0)$, and $\mathcal{Q}' = \mathcal{N}(\Sigma G, \sigma^2)$ for $p(\mathbf{c} \mid \mathbf{y}_0)$. Their KL divergence leads to

$$\nabla D_{\mathrm{KL}}(\mathcal{P}' \,\|\, \mathcal{Q}') \approx \frac{\mathbf{x} - \boldsymbol{\mu}_t - \Sigma G}{\sigma^2}.$$

Hence,

$$\nabla_{\mathbf{x}_t} \log p(\mathbf{c} \mid \mathbf{x}_t, \mathbf{y}_0) = \nabla D_{\mathrm{KL}}(p_\theta(\mathbf{x}_t) \,\|\, p_\theta(\mathbf{c}))$$

is justified.

## 8 ADDITIONAL EXPERIMENTAL RESULT

### 8.1 ADDITIONAL EXPERIMENT OF NUMERICAL ERROR RESTORATION IN REFERENCE REGION

Fig. 10 provides additional experimental proof to support the validness of numerical error restoration by Eq. 11 in reference region.

### 8.2 ADDITIONAL EXPERIMENT OF NUMERICAL ERROR RESTORATION OUT OF REFERENCE REGION

Fig. 11 and Fig. 12 provides additional experimental proof to support the validness of perceptual quality enhancement by Algorithm 1 and Algorithm 2 in non-reference region.

## 9 FUTURE WORK AND EXTENDED GENERALITY

Although our method is not designed as a universal diffusion prior, it offers a complementary choice of generality for frequency-aware and scene-adaptive control. The separation of first- and second-order differential term naturally corresponds to low- and high-frequency behaviors, enabling another controlled enhancement depending on image semantics.

First, certain image categories (e.g., natural scenes) benefit from high-frequency detail enhancement, whereas structured high-frequency domains (e.g., urban patterns) require more conservative correction. Semantic-aware usage policies for perceptual quality enhancement may further stabilize performance in such cases.

Second, mobile wide/tele dual-camera systems provide LR–HR overlap regions that enable per-image scale calibration. This yields practical adaptability and improves generalization to non-overlap regions without additional training.

Third, when HR-like patterns (e.g., text, known symbols) are present, selective reference-guided correction can boost readability and fine-detail recovery, offering benefits for tasks such as real world OCR.

Finally, improving trajectory alignment between LR and HR in posterior space is a promising direction to further enhance fidelity in non-reference regions.

Overall, these directions show that our method enables a different yet practical form of generality—fine-grained, frequency-based, and scene-conditioned—that complements existing diffusion-prior approaches.

## 10 $\nabla D_{KL}$ AS UNSHARP MASKING: A PERCEPTUAL INTERPRETATION

**Connection to classical Laplacian sharpening.** Our KL-based correction step can be interpreted as a variant of classical Laplacian enhancement. In image processing, Laplacian sharpening is often written as

$$\mathbf{I}_{\mathrm{sharp}} = \mathbf{I} - c \, \nabla^2 \mathbf{I}, \quad c > 0,$$

Table 6: Variance comparison of global-wise and image-wise calibration. For each diffusion step, the global-wise method selects a single optimal index across the entire dataset, while the image-wise method estimates an optimal index per an image tile set. Lower variance indicates better alignment between the chosen index and tile-wise minima.

| Diffusion Step | t=0 | t=1 | t=2 | t=3 |
|----------------|-----|-----|-----|-----|
| Global-wise    | 30.6 | 32.2 | 5.4 | 8.6 |
| Image-wise     | 13.0 | 12.7 | 2.2 | 3.7 |

| Global-wise | Image-wise |
|-------------|------------|
| $$m = \arg\min_j \left( \sum_{i=1}^{M} e_i[j] \right)$$ $$\ell_i = \arg\min_j e_i[j]$$ $$\mathrm{Var}_{\mathrm{global}} = \frac{1}{M} \sum_{i=1}^{M} (\ell_i - m)^2$$ | $$m_k = \arg\min_j \left( \sum_{i \in \mathcal{T}_k} e_i[j] \right)$$ $$\ell_i = \arg\min_j e_i[j]$$ $$\mathrm{Var}_k = \frac{1}{|\mathcal{T}_k|} \sum_{i \in \mathcal{T}_k} (\ell_i - m_k)^2$$ $$\mathrm{Var}_{\mathrm{image}} = \frac{1}{N} \sum_{k=1}^{N} \mathrm{Var}_k$$ |

**Variable Definitions:**
$e_i[j]$ : error value of tile $i$ at candidate index $j$;
$j$ : index in the candidate index set (e.g., discretized parameter grid);
$\ell_i$ : tile-wise optimal index for tile $i$;
$m$ : global optimal index selected across the entire dataset;
$m_k$ : optimal index for image $k$ computed from its tile set $\mathcal{T}_k$;
$M$ : total number of tiles in the dataset;
$N$ : number of images;
$\mathcal{T}_k$ : set of tile indices belonging to image $k$;
$\mathrm{Var}_{\mathrm{global}}$ : deviation of tile-wise optima from global optimal index;
$\mathrm{Var}_{\mathrm{image}}$ : averaged deviation for image-wise calibration.

where $\nabla^2 \mathbf{I}$ denotes the discrete Laplacian. Using the standard approximation

$$\nabla^2 \mathbf{I} \approx \mathrm{Smooth}(\mathbf{I}) - \mathbf{I},$$

with $\mathrm{Smooth}(\mathbf{I})$ a low-pass filtered (e.g., Gaussian-blurred) version of $\mathbf{I}$, this becomes

$$\mathbf{I}_{\mathrm{sharp}} \approx \mathbf{I} - c(\mathrm{Smooth}(\mathbf{I}) - \mathbf{I}) = \mathbf{I} + c(\mathbf{I} - \mathrm{Smooth}(\mathbf{I})),$$

which is the familiar unsharp-mask form: the high-frequency component $\mathbf{I} - \mathrm{Smooth}(\mathbf{I})$ is amplified by a positive factor.

Assume two normal distributions $p_{\mathrm{SR}} : \mathcal{N}(\mathbf{x}_{\mathrm{SR}}, \sigma_t^2), p_{\mathrm{LR}} : \mathcal{N}(\mathbf{x}_{\mathrm{LR}}, \sigma_t^2)$. From Lemma 3.2.1, the gradient of the KL divergence between $p_{\mathrm{SR}}$ and $p_{\mathrm{LR}}$ with respect to $\mathbf{x}_{\mathrm{SR}}$ is

$$\nabla_{\mathbf{x}_{\mathrm{SR}}} D_{\mathrm{KL}}(p_{\mathrm{SR}} \| p_{\mathrm{LR}}) \propto \frac{\mathbf{x}_{\mathrm{SR}} - \mathbf{x}_{\mathrm{LR}}}{\sigma_t^2}.$$

If we assume that the LR image is approximately a smoothed version of the SR output,

$$\mathbf{x}_{\mathrm{LR}} \approx \mathrm{Smooth}(\mathbf{x}_{\mathrm{SR}}),$$

then $\mathbf{x}_{\mathrm{SR}} - \mathbf{x}_{\mathrm{LR}}$ corresponds to a high-pass (edge/detail) component, analogous to $\mathbf{I} - \mathrm{Smooth}(\mathbf{I})$ in unsharp masking.

This immediately explains the effect of the sign:

- **Gradient descent (fidelity-oriented).** A standard KL-descent step

$$\mathbf{x}_{\mathrm{new}} = \mathbf{x}_{\mathrm{SR}} - \alpha \nabla_{\mathbf{x}_{\mathrm{SR}}} D_{\mathrm{KL}}(p_{\mathrm{SR}} \| p_{\mathrm{LR}}) = \mathbf{x}_{\mathrm{SR}} + \beta(\mathbf{x}_{\mathrm{LR}} - \mathbf{x}_{\mathrm{SR}}), \quad \beta > 0,$$

moves $\mathbf{x}_{\mathrm{SR}}$ toward the (smoothed) LR image. This reduces high-frequency content (a "blur" direction), thereby improving fidelity metrics (e.g., PSNR/SSIM) at the cost of perceptual sharpness.

- **Sign-flipped step (perception-oriented).** In contrast, the sign-flipped update

$$\mathbf{x}_{\mathrm{new}} = \mathbf{x}_{\mathrm{SR}} + \alpha \, \nabla_{\mathbf{x}_{\mathrm{SR}}} D_{\mathrm{KL}}(p_{\mathrm{SR}} \, \| \, p_{\mathrm{LR}}) = \mathbf{x}_{\mathrm{SR}} + \gamma \, (\mathbf{x}_{\mathrm{SR}} - \mathbf{x}_{\mathrm{LR}}), \quad \gamma > 0,$$

can be written, under $\mathbf{x}_{\mathrm{LR}} \approx \mathrm{Smooth}(\mathbf{x}_{\mathrm{SR}})$, as

$$\mathbf{x}_{\mathrm{new}} \approx \mathbf{x}_{\mathrm{SR}} + \gamma\big(\mathbf{x}_{\mathrm{SR}} - \mathrm{Smooth}(\mathbf{x}_{\mathrm{SR}})\big), \quad \gamma > 0$$

which is exactly an unsharp-mask or Laplacian-style sharpening step that amplifies edges and textures.

Therefore, negative (sign-flipped) $\nabla D_{KL}$ steps naturally move the solution toward sharper, more texture-rich images, which increases perceptual metrics such as LPIPS, DISTS, and CLIPIQA, while positive (descent) steps move toward smoother, LR-like images that favor traditional fidelity metrics.

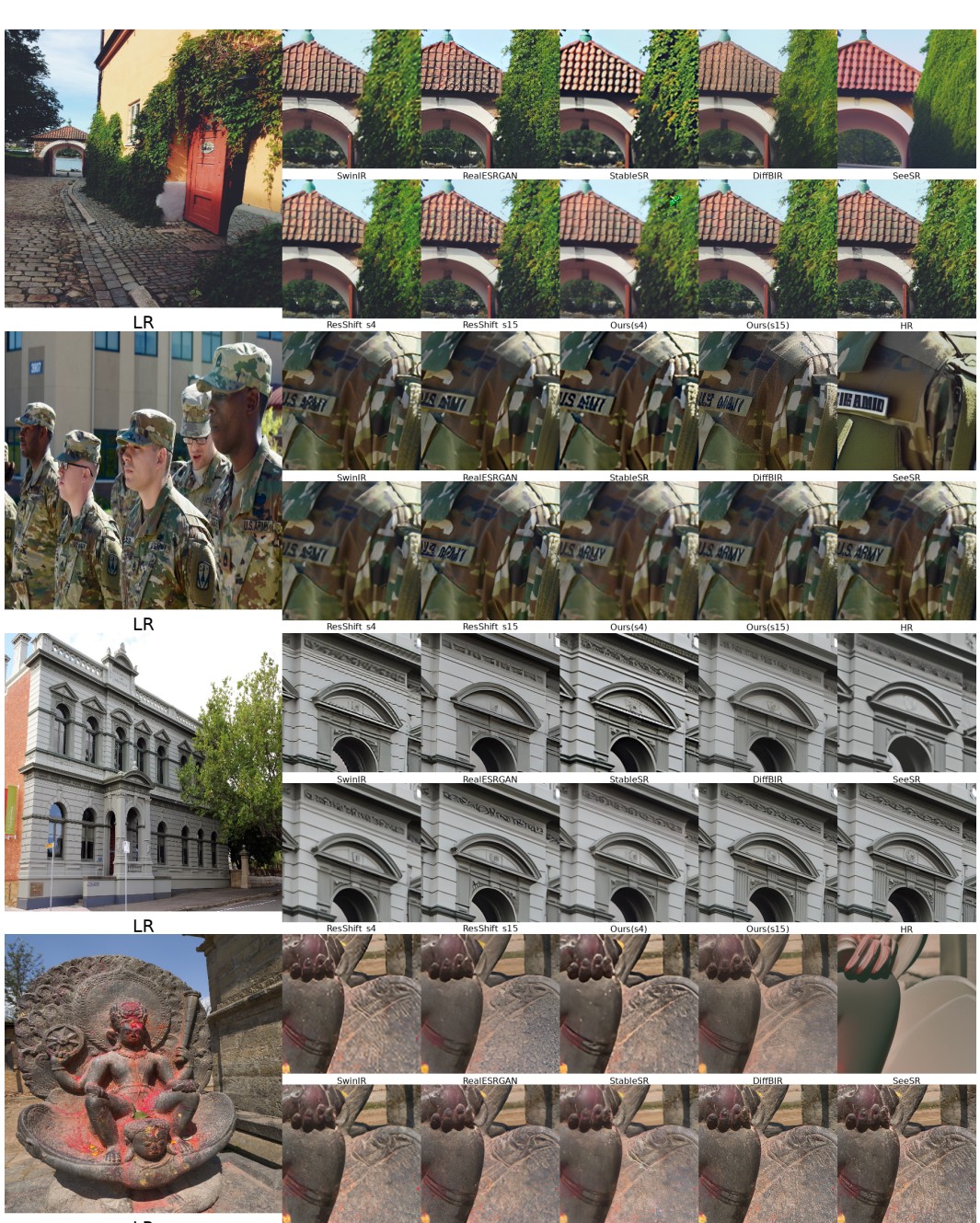

Figure 10: Experiment for Reference region. Qualitative comparison on the Flickr30k dataset Young et al. (2014). Ours (s4) and Ours (s15) represent ResShift (s4 and s15) with the proposed restoration in Eq. 11. Please note that most of the collapsed patterns and text that failed to be reconstructed in the baseline have been successfully restored.

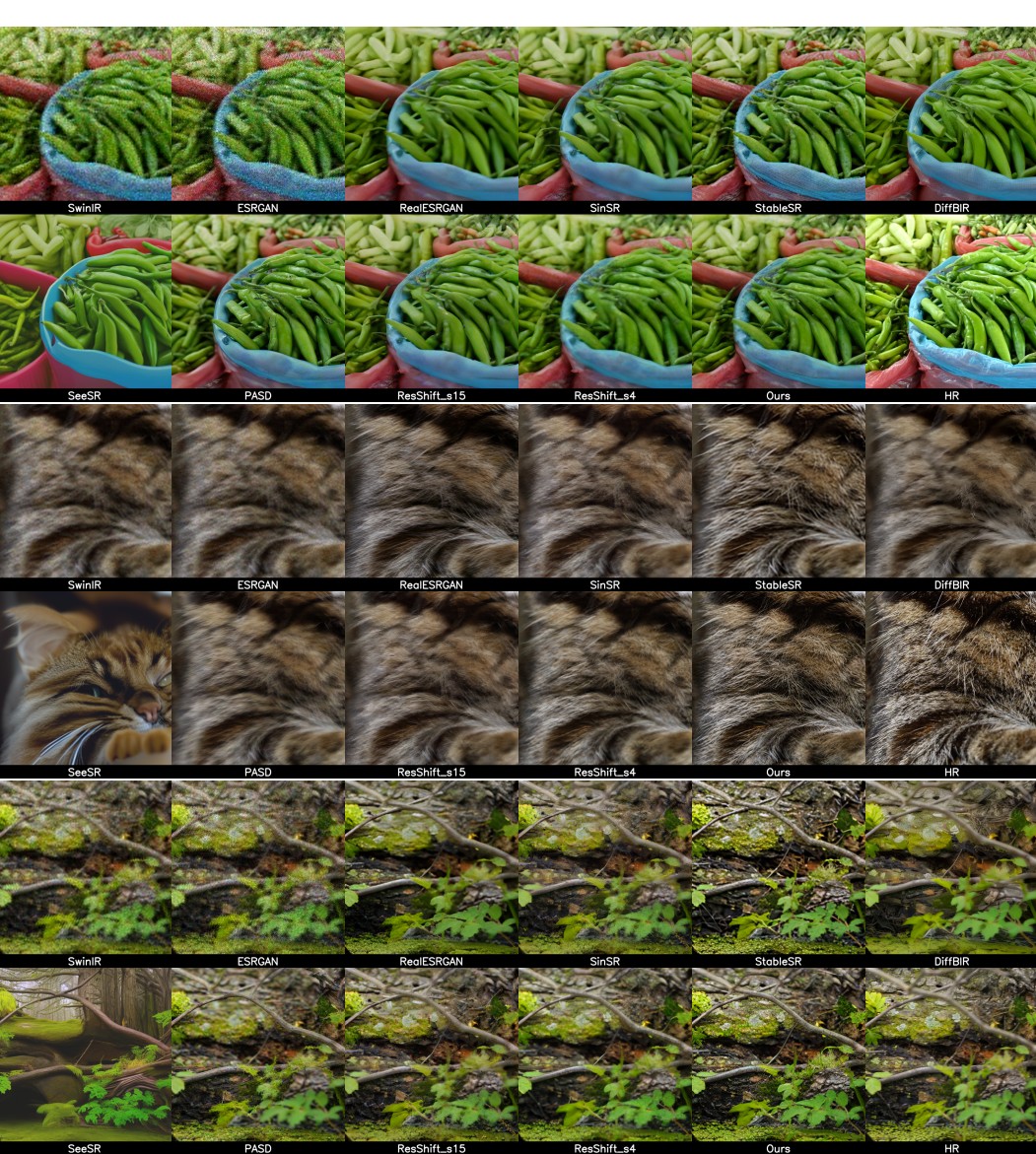

Figure 11: Experiment for Non-Reference region. Qualitative comparisons of image quality across various algorithms on the DIV2K dataset Agustsson & Timofte (2017). Our method demonstrates outstanding performance in enhancing the restoration of unstructured objects that lack clear semantic information. Note the significant perceptual quality enhancement compared to the baseline ResShift network.

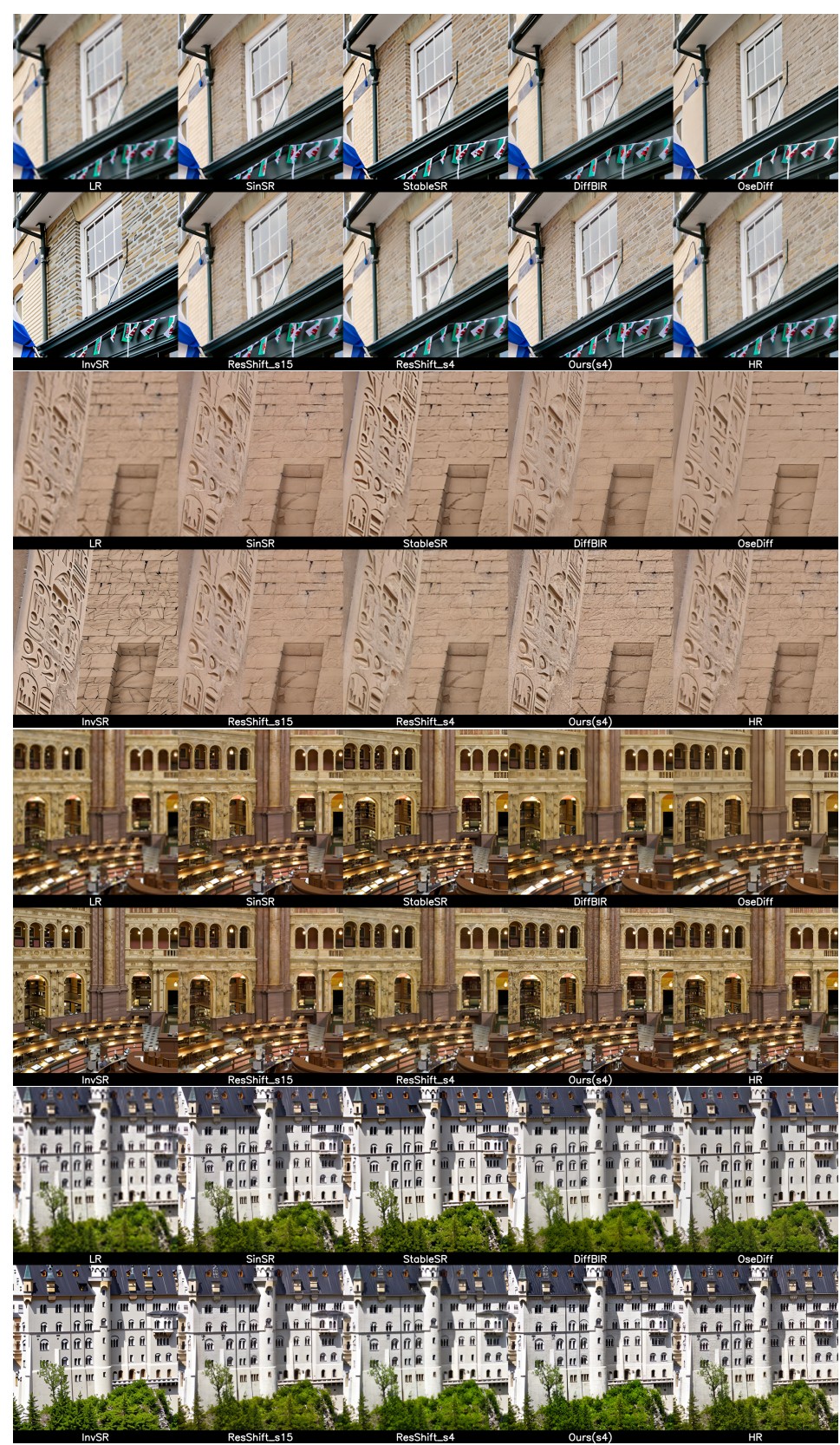

Figure 12: Non-reference region. Visual comparison against state-of-the-art SR methods. Our method produces more details than other competing ones.

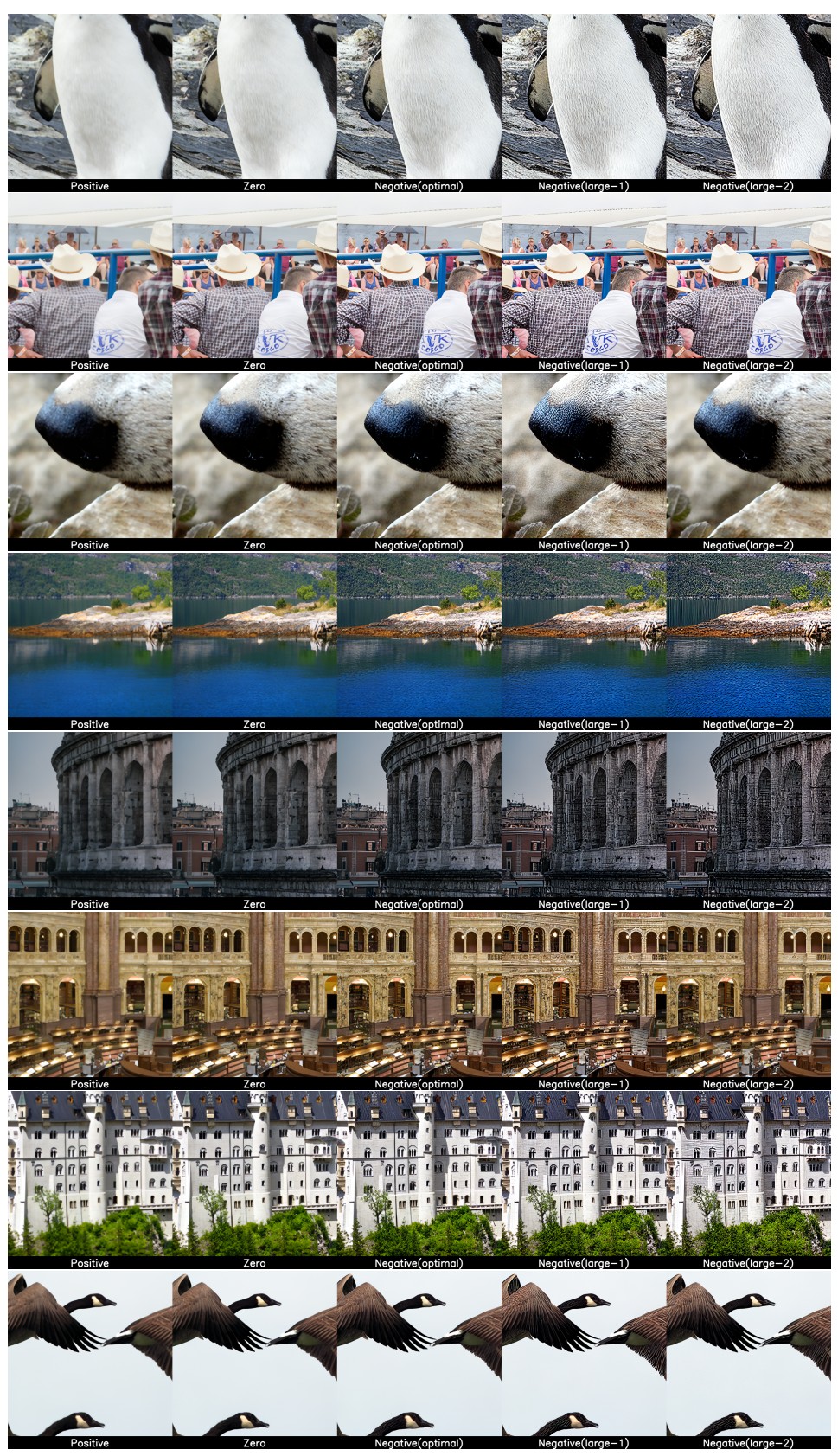

Figure 13: Each parameter set is listed in the table below. The first three columns illustrate the effect of positive, zero, and moderately negative values, while the last two use larger negative magnitudes that lead to distinct artifact patterns. This comparison highlights the sensitivity of the correction module to the sign and absolute scale of the parameters.

| Positive | Zero | Negative(optimal) | Negative(large-1) | Negative(large-2) |
|---|---|---|---|---|
| $(1700, 70, 60)$ | $(0, 0, 0)$ | $(-1700, -70, -60)$ | $(-3000, -100, -100)$ | $(-2200, -140, -120)$ |