# OpenReview forum: "Plug-in Image Quality Control for Posterior Diffusion Super-Resolution"
_ICLR.cc/2026/Conference — Submitted to ICLR 2026_

### Official Review · Reviewer_Zmc6 · 2025-10-24

**Soundness:** 2
**Presentation:** 1
**Contribution:** 1
**Rating:** 2
**Confidence:** 4

**Summary:**

The presented work extends posterior diffusion SR methods (namely, ResShift) by introducing a plug-in image quality control (IQC) mechanism. The proposed correction term is derived from a numerical analysis linking discretization error in the reverse diffusion ODE to the gradient of a KL divergence. This enables post-hoc "quality control" without retraining. Overall, the method provides theoretical interpretability and mild improvements in perceptual quality but limited quantitative gains over ResShift.

**Strengths:**

- Introduces a numerical-analysis perspective on diffusion SR, which is kind of interesting.
- Demonstrates that image quality can be controlled, although the shown example with HR reference is quite impractical (since HR is usually not given), but can be seen as upper bound (although HR itself is the upper bound).

**Weaknesses:**

- The core derivations rely on overly strong Gaussian (for a posterior learning approach) and constant-variance assumptions. I am a bit puzzled by this since ResShift does not need these assumptions and still performs comparable, sometimes better, questioning the goal of this work.
- Quantitative gains over ResShift are marginal (in general mostly mixed), suggesting the proposed correction has little real impact despite its theoretical framing...
- The paper suffers from numerous language and formal issues (e.g., spelling errors such as latant space, misuse of key words like "proof of definition", and inconsistent notations), which reduce readability and professionalism for this type of conference.
- Theorem 3.2.1 as a proof is not rigorous: it stitches pieces with unproven proportionalities and drops constants, so it should be framed as a modeling assumption backed by empirical calibration (i.e., A, B regression), not as a theorem.
-  Comparisons are limited to minor ablations (LR vs. HR guidance) without broader baselines or qualitative failure analysis to support the claimed controllability advantage.

**Questions:**

- In practice, variance is usually not constant spatially (different noise levels per region) nor temporally (it decays along t). In contrast, ResShift does not assume a fixed variance, nor does it linearize everything to one isotropic sigma^2. Therefore, it explicitly learns how the noise level and residual destribution behave given the LR input. Why did the authors try to simplify that? Did I miss something major here?
- In Theorem 3.2.1 (spatial derivative), the proof equates (by using the time derivative from proposition 3.2.1) temporal derivatives of the drift with spatial gradients of the KL divergence. What formal justification supports this operator equivalence?
- Given that the IQC model builds directly on ResShift with a simple correction term, how does this constitute a novelty?
- Since HR ground truth is unavailable at inference (in practice), what practical insight does "Ours (HR)" provide beyond an upper bound? Is the upper bound not already given by the HR image itself? It is a bit deceiving with the (high) reported numbers in combination with the words "ours", although it seems to be an upper bound...

---

> ### Author Response · Authors · 2025-11-26
>
> We thank the reviewer for the constructive comments and address each point below.
>
> **1. Constant variance assumption**
>
> Thank you for pointing this out. We identified a notation mistake in Assumption 2 and have corrected it in the revised PDF. Our method follows a temporally dependent and spatially fixed predefined variance schedule, which aligns with standard practices in pretrained model of resshift.
>
> **2. Marginal quantitative gain**
>
> Thank you for the observation. The mixed quantitative gains are expected because our method follows the well-known perception–distortion tradeoff discussed in ResShift. Improving perceptual quality inherently degrades fidelity, and vice versa. Our contribution is that the proposed correction enables continuous control along this tradeoff, not simultaneous improvement of all metrics. Therefore, Table 2 reports a balanced operating point.
> Figure 13 illustrates artifacts when the parameter magnitude becomes too large. Achieving strong perceptual enhancement with minimum artifacts would require a multi-camera setup as in Figure 7(b), which we leave as future work.
>
> **3. Numerous language and formal issues**
>
> We appreciate you for pointing this out. we revised all issues.
>
> **4. Temporal–Spatial Derivative Equivalence, Theorem 3.2.1 as a proof is not rigorous**
>
>  We appreciate you for pointing this out.  We clarify: This is not an "operator equivalence" but a structural correspondence arising from the Gaussian Fokker-Planck dynamics. We changed it to the proportionality operator. We also changed Theorem 3.2.1 to Proposition 3.2.2. Pdf is revised accordingly.
>
> **5. Minor ablations and comparisons**
> Revised pdf contains more results.
>
> **6. Novelty: The correction term appears too simple.**
> We understand that the final formula may appear simple, but the conceptual contribution is more substantial.
> In summary, our novelty consists of:
>
> 1. **A plug-in correction module specifically designed for posterior-learning SR models.**
> 2. **Continuous image quality control** enabled through this plug-in module.
> 3. **A principled approach to scale-factor estimation and optimization**, whereas prior work—such as classifier guidance—relies almost entirely on empirical tuning.  We also provide an example of **online estimation** in Figure 7(b).
>
> In conclusion, the core contribution is the principled unification of disparate theoretical elements into a coherent framework for controllable image quality in posterior SR. By establishing the connection between numerical error correction and KL-guided generation, we transform posterior diffusion models from black-box samplers into interpretable, user-controllable systems with explicit fidelity-perception trade-offs.
>
> If the reviewer is aware of prior work addressing similar problems, we would greatly appreciate a reference.
>
> **7. Practical meaning of “Ours (HR)” when HR is unavailable at inference**
> We agree that “Ours (HR)” represents an upper bound, and we state this explicitly in the revision.
> We further describe realistic scenarios where approximate-HR signals can be obtained:
> multi-camera calibration (tele–wide overlap) as in Figure 7(b);
> predefined high-resolution symbols (e.g., medical scans, cartoons);
> similar to real-world practices in RefVSR (Lee et al., link included).
>
> If we have overlooked any part of your question or if you have further concerns, we would be happy to clarify.

---

> > ### Comment · Reviewer_Zmc6 · 2025-11-27
> >
> > I thank the authors for the revised version, as the work is now in noticeably better shape. I will raise my score to 4, however, I am still not entirely convinced by the Gaussian assumption and Spatially Constant Variance, especially for a posterior learning approach. Assuming that the posterior distribution is Gaussian is, in my opinion, a far-fetching stretch. The same goes for the constant variance assumtion...

---

> ### Author Response · Authors · 2025-11-28
>
> We thank the reviewer again for the detailed follow-up and for raising the score.
>
> Regarding the concern that our derivations “rely on overly strong Gaussian and constant-variance assumptions” and that “ResShift does not need these assumptions,” we would like to clarify that ***we do not introduce any additional assumptions beyond ResShift.***
>
> In fact, ResShift itself already adopts an isotropic Gaussian structure per diffusion step. For example, Eqs. (11)–(13) in ResShift explicitly use
> $$
> x_t = x_{t-1} + \alpha_t e_0 + \kappa\sqrt{\alpha_t}\xi_t,
> \quad \xi_t \sim \mathcal N(0, I),
> $$
> which induces the forward transition
> $$
> q(x_t \mid x_{t-1}, e_0)
> = \mathcal N\big(x_{t-1} + \alpha_t e_0\; \sigma_t^2 I \big),
> \quad \sigma_t^2 = \kappa^2 \alpha_t.
> $$
> Similarly, the derivation of Eq. (6) (Eqs. (15)–(17)) shows that the reverse transition can be written as
> $$
> q(x_{t-1} \mid x_t, x_0, y_0)
> = \mathcal N\big(\mu_t^{-}\, \lambda_t^2 I\big)
> $$
> for some mean $\mu_t^{-}$ and scalar variance $\lambda_t^2=\kappa^2 \frac{\eta_{t-1}}{\eta_t}$ and $\alpha_t=\eta_t-\eta_{t-1}$. Thus, at each timestep $t$, both the forward and backward transitions in ResShift are Gaussian with a **scalar, spatially constant (isotropic) covariance** at that step, while the variance level $\sigma_t^2$ (and $\lambda_t^2$) still changes over time via the noise schedule.
>
> In our analysis, we approximate the marginal at step $t$ by
> $$
> q(x_t \mid x_0, y_0)
> \approx \mathcal N\big(\mu_t(x_0, y_0)\, \sigma_t^2 I\big),
> $$
> where $\mu_t(x_0, y_0)$ corresponds to the analytic mean derived in ResShift and $\sigma_t^2$ is the same scalar variance predefined by the per-timestep noise schedule(`def get_named_eta_schedule()`). We do **not** collapse all timesteps into a single global $\sigma^2$; we only assume, as in ResShift, that the covariance at each step is isotropic $\sigma_t^2 I$.
>
> The posterior variance at step $t$ is taken from the precomputed scalar schedule (`self.posterior_variance[t]`) and injected via `_extract_into_tensor(…, x_t.shape)`, which simply broadcasts a **per-timestep scalar** to all spatial locations. Thus, in this configuration ResShift already uses a per-step isotropic Gaussian parameterization; it does *not* explicitly learn spatially non-uniform noise levels or a heteroscedastic distribution conditioned on the LR input.
>
> In conclusion, ***we do not introduce any additional assumptions beyond ResShift.***

---

### Official Review · Reviewer_ReCH · 2025-10-31

**Soundness:** 4
**Presentation:** 3
**Contribution:** 3
**Rating:** 6
**Confidence:** 3

**Summary:**

This paper introduces a novel plug-and-play framework for enhancing the performance of posterior diffusion models in image super-resolution (SR), avoiding the need for an explicit degradation model.
The primary contributions are threefold:
Theoretical Innovation: The authors provide a numerical analysis proving that the discretization error inherent in first-order ODE solvers can be equivalently expressed as the gradient of KL divergence. The main idea unifies numerical error correction with image-based classifier guidance theoretically.
Plug-and-Play Practical Design: Based on the theory, the method incorporates a correction term into the reverse sampling process of a pre-trained model without requiring any retraining. This is implemented via a dual-conditioning mechanism: one path for the noisy LR input and a separate, clean path for an additional guidance image.
Adaptation for Blind SR: They calibrate the guidance scale using a dataset of LR-HR pairs and then apply it using only the Low-Resolution (LR) image during inference (Algorithm 1 & 2), effectively trading off fidelity for enhanced perceptual quality.
Experiments on benchmarks like DIV2K and RealSR show that the method, when plugged into models like ResShift, achieves state-of-the-art performance in no-reference metrics (e.g., MANIQA, CLIPIQA, FID) and perception-based metrics (e.g., LPIPS), with minimal degradation in fidelity metrics (PSNR, SSIM).

**Strengths:**

1.	The paper establishes a principled, mathematical link between numerical integration error and probabilistic divergence. This provides a rigorous foundation for the proposed method and a new analytical tool for the community. The "plug-and-play" nature of the framework is practical for deployment.
2.	The paper validates quantitative comparisons against SOTA methods (regression-based, GAN-based, prior- and posterior-based diffusion models). The results demonstrate comparable performance with other SOTA methods.
3.	This work is interesting that it identifies the underexplored limitations of posterior diffusion SR (discretization errors, lack of control), making its contributions highly relevant.

**Weaknesses:**

1.	The calibration of the guidance scale relies on a set of LR-HR image pairs. This dependency on HR data for a "blind" SR method is a limitation, as the optimal scale might be dataset-dependent and may not generalize perfectly to all real-world degradations. The results on RealSR and DIV2K also reveal this weakness, the PSNR index is weak compare to ResShift with same inference step.
2.	The theoretical derivation relies on a constant variance assumption (Assumption 2) and linearizes the relationship between error and the KL gradient. The practical impact of these simplifying assumptions on complex, real-world image distributions is not fully quantified.
3.	Limited Comparison with Prior-based Guidance. The discussion focuses on posterior models. A more direct comparison or discussion on how this guidance principle relates to or differs from guidance mechanisms commonly used in prior-based models (e.g., using CLIP) would be better.

**Questions:**

Refer to the weaknesses. Robustness of Parameter Calibration:How sensitive is the method's performance to the specific dataset used for calibrating the guidance scale? Have the authors tested the generalization of a scale calibrated on one dataset (e.g., ImageNet) when applied to images from a different domain?

---

> ### Author Response · Authors · 2025-11-26
>
> We thank the reviewer for the constructive comments and address each point below.
>
> **1. Robust calibration**
>
> Thank you for the thoughtful question. We have substantially revised the experimental section to clarify the calibration strategy and its limitations. Please refer to Supplementary Section 6.1 (updated calibration strategy), Figure 6, and Tables 3, 4, and 6.
>
> The reviewer is correct: the robustness of parameter calibration is a practical limitation. Recovering an optimal scale *must* be solved carefully in practice, and the original version briefly described this to avoid distracting from the theoretical focus. The revised version now provides:
>
> - **Figure 6:** a detailed stability analysis showing that estimation on a tile set of an image  or on a specific domain yields *lower variance* than using a single globally calibrated scale;
> - **Figure 7(b):** a plug-and-play example where tele-camera–calibrated parameters are applied to all regions using wide/tele overlap;
> - **Table 6:** the quantitative comparison showing the discrepancy between globally calibrated versus online-estimated parameters on a tile set of an image.
>
> Regarding generalization:
> The guidance scale is fundamentally **dataset- and camera-dependent**, and ideally should be re-estimated per image. In most practical settings this is not feasible, but in structured environments (e.g., multi-camera systems as in Figure 7(b)) per-image or per-region calibration *is* possible. The revised version now explicitly discusses this limitation and shows failure cases arising from miscalibration (Figures 6 and 13).
>
> More importantly, our method obeys the same **perception–distortion tradeoff** described in ResShift. Enhancing perceptual quality necessarily degrades fidelity, and vice versa. The updated experiment section explains this through the natural metric ordering:
>
> PSNR/SSIM: SR->bicubic(LR)->HR
>
> LPIPS/DISTS/NIQE: bicubic(LR)->SR->HR
>
> MANIQA/CLIPQA: bicubic(LR)->HR->SR
>
> - **Positive correction** minimizes SR–HR loss → higher fidelity, lower perceptual quality.
> - **Negative correction** increases the loss relative to HR → higher perceptual quality, lower fidelity.
>
> To support this theoretically, we added **Proposition 3.2.3**, showing that both the SR–HR gradient and the SR–LR gradient can serve as valid correction directions.
> Figure 5 analyzes the trajectory mismatch between these gradients, while Algorithms 1 and 2 provide a principled parameter-estimation strategy. The stability of this estimation method is examined in Figure 6.
>
> In summary, while calibration is indeed a limitation—and we appreciate the reviewer pointing this out—the revised manuscript now provides both (a) a clearer explanation of its sensitivity and (b) concrete mechanisms for online, plug-and-play calibration where feasible.
>
> **2. Constant variance**
>
> We apologize for the confusion caused by a notation mistake in the original version.
> This has been corrected in the revised PDF.
>
> To clarify: **we do not assume a temporally constant variance.**
> Our method follows the standard diffusion setting in which the **variance changes at every diffusion step**. (i.e., it is *temporally varying*) Thus, Assumption 2 does not impose a globally constant variance over all steps, and the linearization is performed under the same variance schedule at each diffusion step.
>
> **3. Diffusion prior guidance vs posterior guidance**
>
> **Diffusion Prior–based SR** methods (e.g., ControlNet, CLIP-guided SR, StableSR) keep the  pretrained diffusion prior $p(x)$ fixed and introduce a *new conditional network* (ControlNet branch, CLIP encoder, or LoRA modules) to approximate the likelihood term $p(y \mid x)$.  In these methods, the SR model injects LR information into the prior via learned conditioning, and the prior score $\nabla_x \log p(x)$ is modified indirectly through these auxiliary modules.
>
> **Posterior-learning SR** (e.g., ResShift ) follows a different philosophy: it directly models or corrects the posterior model $ \nabla_x \log p(x_t \mid y)$, treating SR as an inverse problem. Because the likelihood is covered via dataset, posterior-based SR requires no new conditioning network, and image quality can be controlled by adjusting the posterior dynamics in our method.
>
> Due to page limit in the main page and organization in the supplemental material, I still do not find where it can be added. I will add it as soon as I find it.
>
> If we have overlooked any part of your question or if you have further concerns, we would be happy to clarify.

---

### Official Review · Reviewer_zA8j · 2025-11-02

**Soundness:** 2
**Presentation:** 3
**Contribution:** 3
**Rating:** 4
**Confidence:** 4

**Summary:**

This paper proposes a plug-in module for posterior diffusion super-resolution models to control image quality. The authors establish a theoretical link between the numerical discretization error in the sampling process and the gradient of the KL divergence. Based on this, they propose an image-based guidance method. In principle, this method can use a high-resolution (HR) reference to correct errors and improve fidelity. In practice, where no HR reference is available, the paper suggests flipping the sign of the guidance term (using the low-resolution image as a guide) to enhance perceptual quality without retraining the base model.

**Strengths:**

The main strength of this paper lies in its novel theoretical insight. Connecting the numerical discretization error of the ODE solver to a probabilistic concept like the KL divergence gradient is elegant and provides a new perspective on analyzing diffusion model trajectories. The idea of unifying numerical error correction with classifier guidance is intriguing. The experiments do show that the proposed method can improve perceptual metrics, which is a valuable goal for super-resolution.

**Weaknesses:**

While the theoretical premise is interesting, I have several major reservations about the work in its current form.

My primary concern is the conceptual leap between the paper's theory and its practical application. The theory elegantly justifies correcting numerical error to improve fidelity by moving the generation towards an HR target. However, the proposed blind SR method does the exact opposite: it pushes the generation away from the LR condition to ostensibly improve perception. This sign-flipping feels less like a principled application of the core theory and more like a clever but ad-hoc heuristic that happens to yield perceptually pleasing results. The paper lacks a convincing theoretical argument for why this maneuver should work consistently.

This leads to my second point regarding the "plug-and-play" and "no-training" claims, which I find to be overstated. The method's reliance on a calibration step (Algorithm 1) using a set of LR-HR pairs is a form of data-driven parameter tuning. This undermines the "no-training" narrative and raises questions about generalization. A single guidance scale A calibrated on one dataset may not be robust across diverse image domains.

Furthermore, the paper does not sufficiently address the inherent risks of its approach. Pushing the generation away from the LR input is a fundamentally unstable process that could easily introduce artifacts or hallucinate details inconsistent with the source. A thorough analysis of these failure modes is conspicuously absent. Finally, the empirical validation is quite narrow. Claiming a general framework for "posterior diffusion SR" while testing on only a single backbone model (ResShift) weakens the paper's claims of broad applicability.

**Questions:**

The sign-flipped guidance is the linchpin of your practical method, yet its theoretical grounding seems to be the weakest part of the paper. Could you elaborate on the justification for this? Specifically, what prevents this "push away from LR" from simply amplifying noise or creating implausible structures, rather than consistently enhancing perceptual quality?

The "plug-and-play" claim hinges on the robustness of the calibrated scale A. How does the method's performance hold up when A is calibrated on one domain (e.g., natural images) and tested on a completely different one (e.g., medical scans, cartoons)? This is crucial for understanding the method's true generalizability.

A complete picture of an algorithm's performance includes its limitations. Could you provide a qualitative analysis of failure cases? It would be particularly insightful to see instances where the perceptual enhancement introduces clear artifacts or generates textures that are inconsistent with the LR input, as this would help clarify the method's trade-offs.

I am open to reconsidering my evaluation should the authors provide convincing answers to these questions in their rebuttal.

---

> ### Author Response · Authors · 2025-11-26
>
> **1. Conceptual leap**
> Thank you for the insightful comment. We have updated both the qualitative and quantitative analyses in the experiment section to clarify this behavior. In summary, different metrics naturally obey the following ordering:
>
> PSNR/SSIM: SR->bicubic(LR)->HR
>
> LPIPS/DISTS/NIQE: bicubic(LR)->SR->HR
>
> MANIQA/CLIPQA: bicubic(LR)->HR->SR
>
> Because of this phenomenon, using a **positive correction** moves the solution toward the HR target, improving fidelity but reducing perceptual quality. Conversely, a **negative correction** increases the loss relative to HR and moves the solution in the opposite direction, thereby enhancing perceptual quality at the cost of fidelity.
>
> To anchor this more firmly in theory, we added **Proposition 3.2.3**, which shows that not only the SR–HR gradient but also the SR–LR gradient can serve as a valid correction direction. Figure 5 analyzes the trajectory mismatch between these two gradients, and Algorithms 1–2 provide a principled parameter-estimation mechanism to address this issue. Stability properties of the estimation procedure are further examined in Figure 6.
>
> These additions clarify why sign flipping is not an ad-hoc trick but a consistent consequence of the underlying gradient structure and the perception–distortion tradeoff.
>
> **2. Plug-and-play, no training claim**
>
> Thank you for raising this point. We intentionally kept the discussion on the practical model lightweight to avoid distracting from the theoretical focus, but we have expanded this section in the revised version.
>
> We agree that the procedure used to produce the results in Table 2 is not “plug-and-play” in the strictest sense. However, it differs substantially from traditional training, as we uses only a small set of approximately 500 LR–HR pairs for calibration rather than full-scale optimization and just one computation for a set.
>
> Importantly, **Algorithms 1 and 2** provide a mechanism for truly plug-and-play operation in scenarios where online calibration is feasible. As illustrated in **Figure 7(b)**, multi-camera systems with overlapping wide/tele regions naturally provide such calibration environment at inference time.
>
> Regarding robustness across diverse image domains, **Figure 6** analyzes the stability of the parameter estimation, and the multi-camera example in Figure 7(b) again demonstrates how the proposed system can generalize when appropriate domain-specific signals are available.
>
> **3. Usages on other domains**
> Thank you for the important question. The behavior of **Ours(LR,–)**—which enhances perceptual detail by moving away from the LR condition—is particularly suited for natural scenes where additional high-frequency content is visually plausible. In contrast, domains such as medical scans or cartoons require **strict structural accuracy**, and blindly adding detail would not be appropriate.
>
> As discussed in the *Future Works* section, if a domain provides **predefined high-resolution symbols or reference structures**, these can serve as substitutes for HR supervision, enabling the use of **Ours(HR,+)** instead. In fact, domains like medical imaging and cartoons are promising candidates for this approach because they often contain well-defined canonical structures that can act as reliable HR proxies.
>
> **4. Limited test on a single platform**
>
> We appreciate the reviewer’s concern. At present, only a single posterior-learning SR model (ResShift) exists in the literature, so evaluating our framework on one backbone is unfortunately unavoidable. However, our results demonstrate that posterior models are also capable of **continuous image quality control**, which opens the door for a wider family of posterior-based SR models—much like the diversity observed in diffusion prior methods. We discuss these research opportunities in the *Future Works* section.
>
> Regarding instability and failure modes: we agree that pushing the generation away from the LR input can introduce artifacts. This is precisely why we analyze parameter stability in Figure 6 and examine failure cases (e.g., grain artifacts under large corrections) in the revised qualitative section. The framework therefore not only acknowledges these risks but provides explicit diagnostics for understanding and mitigating them.
>
> **4. Performance limitations**
> Thank you for the suggestion. As shown in the 4th and 5th examples of Figure 13, when the parameter value moves outside the optimal operating range, noticeable failures such as grain artifacts begin to appear. We also observe that the optimal parameter range varies across diffusion steps, which highlights the importance of an **online estimation strategy** rather than relying on a fixed, globally tuned value.
>
> If we have overlooked any part of your question or if you have further concerns, we would be happy to clarify.

---

> > ### Comment · Reviewer_zA8j · 2025-11-26
> >
> > I appreciate the authors' huge effort in the rebuttal. The updates show your dedication. However, your response actually confirms my concerns regarding the method's fundamental limitations.
> >
> > 1. Seesaw vs. Slide   As a researcher in this field, I know the fidelity-perception trade-off well. It is a "seesaw" problem we all face. Most research tries to lift the seesaw (improve the upper bound) via data, losses, or degradation modeling.Your response, however, suggests you just built a slide on the existing seesaw. You allow users to slide towards "perception" by sacrificing fidelity, like CFG in generative tasks. Unlike them, SR is very sensitive to distortion. Your "sign-flipping" strategy is essentially a heuristic that risks introducing artifacts, which you acknowledged in the failure cases. This looks like navigating a limitation, not solving it.
> >
> > 2. Calibration and Robustness   On the calibration of parameter $A$, you admitted that global parameters are inferior to image-specific ones. This confirms that robustness is a major issue. If a method requires tuning per dataset (or even per image) to work well, the "training-free" claim loses its practical value. In this context, is a heuristic method really better than simply training on large-scale data to ensure generalization?
> >
> > 3. Theoretical Maturity   The perspective of linking numerical error to KL gradient is novel. However, Reviewer Zmc6's comments forced you to make significant fixes to the method (downgrading theorems to propositions, fixing notations). This makes me doubt if the theory and initial experiments were rigorous enough in the original submission.
> >
> > I recognize the novelty of the proposed angle. I have read your response and do not intend to request further heavy experiments or proofs. I will hold my reservation for now and keep an eye on the discussion with other reviewers.

---

> ### Author Response · Authors · 2025-11-28
>
> ### **Q1 (Seesaw vs. Slide)**
>
> We appreciate the reviewer's clear seesaw/slide analogy and agree that simultaneously improving both fidelity and perception is the ultimate goal.
>
>
> 1. **Current single-image diffusion SR landscape**
>
> For *single-image* diffusion SR, we are not aware of methods that robustly improve both fidelity and perception over strong baselines across multiple datasets. In our own evaluation (Table 2), we consistently observe a trade-off:
>
> - Methods such as StableSR, SeeSR, PASD, OseDiff, and InvSR primarily **enhance perceptual metrics** while sacrificing fidelity.
> - DiffBIR exposes a control parameter $s \in [0,1]$ that trades fidelity for perception, but is **extremely slow** in practice (Table 5), making it difficult to deploy under realistic low-latency constraints.
>
> Some works arguably claim to move the frontier slightly, though robust cross-dataset validation remains limited. To the best of our knowledge, simultaneous improvements in both metrics are more commonly observed in **Multi-Frame SR** approaches (e.g., RefVSR, DBSR).
>
>
> 2. **What we mean by "solving" vs. "navigating" a limitation**
>
> - ResShift and other SR methods require **manual tuning or heuristic exploration** of guidance parameters (e.g., variance schedules, conditioning strength) when moving along the fidelity-perception trade-off.
> - In contrast, we derive a **KL-gradient-based correction direction** from a theoretical analysis of low-step discretization error and use it to **systematically evaluate and select parameters** (e.g., Fig. 13, where operating points are determined via our formulation rather than brute-force search).
>
> In other words, our method does not merely enable arbitrary navigation; it provides a **mathematically grounded criterion for selecting operating points** along the fidelity-perception curve. From our perspective, this is closer to "solving a parameter selection problem under a given constraint" than to simple navigation.
>
>
>
> 3. **On the "sign-flipping heuristic" and artifacts**
>
>
> - **sign-flipping and classical Laplacian sharpening.**
> Our KL-based correction step can be interpreted as a variant of classical Laplacian enhancement. In image processing, Laplacian sharpening is often written as
> $$
> I_{\text{sharp}} = I - c  \nabla^2 I, \quad c > 0,
> $$
> where $\nabla^2 I$ denotes the discrete Laplacian. Using the standard approximation
> $$
> \nabla^2 I \approx \text{Smooth}(I) - I,
> $$
> with $\text{Smooth}(I)$ a low-pass filtered (e.g., Gaussian-blurred) version of $I$, this becomes
> $$
> I_{\text{sharp}}
> \approx I - c (\text{Smooth}(I) - I)
> = I + c (I - \text{Smooth}(I)),
> $$
> which is the familiar unsharp-mask form: the high-frequency component $I - \text{Smooth}(I)$ is amplified by a positive factor.
> Assume two normal distributions
> $p_{\mathrm{SR}} : \mathcal{N}(x_{\mathrm{SR}}, \sigma_t^2)$ and
> $p_{\mathrm{LR}} : \mathcal{N}(x_{\mathrm{LR}}, \sigma_t^2)$.
> From Lemma 3.2.1, the gradient of the KL divergence between
> $p_{\mathrm{SR}}$ and $p_{\mathrm{LR}}$ with respect to $x_{\mathrm{SR}}$ is
> $$
> \nabla_{x_{\text{SR}}} D_{\mathrm{KL}}(p_{\text{SR}}   \|   p_{\text{LR}})
> \propto  \frac{x_{\text{SR}} - x_{\text{LR}}}{\sigma_t ^2}
> $$
> If we assume that the LR image is approximately a smoothed version of the SR output,
> $$
> x_{\text{LR}} \approx \text{Smooth}(x_{\text{SR}}),
> $$
> then $x_{\text{SR}} - x_{\text{LR}}$ corresponds to a high-pass (edge/detail) component, analogous to $I - \text{Smooth}(I)$ in unsharp masking.
> This immediately explains the effect of the sign of our update:
>
> - **Gradient descent (fidelity-oriented).**
>   A standard KL-descent step
>   $$
>   x_{\text{new}}
>   = x_{\text{SR}} - \alpha \nabla_{x_{\text{SR}}} D_{\mathrm{KL}}(p_{\text{SR}} \|  p_{\text{LR}})
>   = x_{\text{SR}} + \beta  (x_{\text{LR}} - x_{\text{SR}}),
>   \quad \beta > 0,
>   $$
>   moves $x_{\text{SR}}$ toward the (smoothed) LR image. This reduces high-frequency content (a “blur” direction), thereby improving fidelity metrics (e.g., PSNR/SSIM) at the cost of perceptual sharpness.
>
> - **Sign-flipped step (perception-oriented).**
>   In contrast, the sign-flipped update
>   $$
>   x_{\text{new}}
>   = x_{\text{SR}} + \alpha \nabla_{x_{\text{SR}}} D_{\mathrm{KL}}(p_{\text{SR}} \| p_{\text{LR}})
>   = x_{\text{SR}} + \gamma (x_{\text{SR}} - x_{\text{LR}}),
>   \quad \gamma > 0,
>   $$
>   can be written, under $x_{\text{LR}} \approx \text{Smooth}(x_{\text{SR}})$, as
>   $$
>   x_{\text{new}}
>   \approx x_{\text{SR}} + \gamma \bigl(x_{\text{SR}} - \text{Smooth}(x_{\text{SR}})\bigr),
>   $$
>   which is exactly an unsharp-mask or Laplacian-style sharpening step that amplifies edges and textures.
>
> Therefore, negative (sign-flipped) KL-gradient steps naturally move the solution toward sharper, more texture-rich images, which increases perceptual metrics such as LPIPS, DISTS, and CLIPIQA, while positive (descent) steps move toward smoother, LR-like images that favor traditional fidelity metrics. This is updated as section 10.

---

> ### Author Response · Authors · 2025-11-28
>
> 4. **Closed-form optimum vs. controllability and practical constraints**
>
>
> - **Our method does not aim to replace large-scale training-based approaches**, but rather addresses some of their practical limitations.
> - Different stakeholders want **different operating points**: professional users (e.g., photographers, scientific imaging) tend to prioritize fidelity, while general consumers often prefer stronger perceptual enhancement even with minor artifacts.
> - A fixed image quality determined purely by training is **difficult to adjust once the model is deployed**. If the pretrained compromise does not satisfy a specific client or validation team, the main recourse is expensive retraining.
> - Our framework is designed for this reality: it allows **fast post-hoc adjustment** of the operating point (via a small number of guidance parameters informed by our KL-based theory) without modifying the underlying model weights.
>
> One could also incorporate semantic priors on top of our KL-gradient framework (e.g., combining $\nabla D_{\mathrm{KL}}(p(x)  \|  p(\text{LR}))$ with CLIP or ControlNet), but even then, failure cases would be determined by the chosen prior, not eliminated.
>
> From a product perspective, especially in camera pipelines involving AWB, RAW processing, and post-processing stages—where validation teams repeatedly refine the target look—this controllability is not merely “a slide”; **it is an essential feature**. This is not a niche scenario: major smartphone manufacturers routinely adjust imaging parameters across product lines and regions to match local preferences, without retraining their entire pipeline. Our framework enables similar flexibility for the SR component.
>
>
>
>
> ### **Q2 (Calibration and Robustness).**
>
> 1. **Calibration overhead vs. training cost**
>
> Our calibration overhead is on the order of **minutes**, whereas large-scale SR training is typically on the order of **days**.
> -We estimate that training of ResShift, DiffBIR, and SeeSR  usually requires **at least 3 days** on a single H100 GPU and training of distillation-style models such as OSEDiff still require **1 day or more**.
> - In contrast, the calibration used to produce Table 2 takes **15 minutes** on a single H100 GPU.
> -Per-image calibration in multi-camera scenarios such as Fig. 7(b) costs about **approximately 50% of total inference time**.
> - Once a calibration is obtained, they can be reused to adjust image quality continously without additional recalibration.
>
> Thus, the overhead of calibration is **well below 1%** of full retraining. In this sense, we used the word "training-free” relative to large-scale retraining.  To be more precise,  we have  revised the terminology from *plug-and-play* to *plug-in module* throughout the paper.
>
> 2. **Robustness, generalization, and the role of our method**
>
> Traditional SR solutions cannot adjust the learned image quality once training is finished. Our framework, in contrast, does **not** require that the calibrated parameters be fixed or always used; it instead provides **flexible control** over image quality at inference time.
> We agree that, in terms of **robust generalization**, large-scale trained models are superior. Our goal is **not** to replace large-scale training, but to **complement** it:
> - Even after training on one million images for several days on an H100 GPU, there is no guarantee that stakeholders will be satisfied with the resulting image quality.
> - When stakeholders request “more detail” or “less detail,” practitioners still resort to **heuristic** and **empirical** parameter tuning (e.g., the manual adjustment of hyperparameters in ResShift, Fig. 3 of its paper), rather than a theoretically guaranteed recipe.
> In practice, if a 3-day training run is rejected by stakeholders , retraining from scratch costs **another days**, with no guarantee that the next result will be accepted. Repeating this loop multiple times easily stretches over **several weeks**.
> By contrast, our method allows the practitioner to generate **many different image-quality variants within a single day** within calibrated limitations and collect feedback quickly. In a real imaging pipeline, SR interacts with RAW processing, post-processing, AWB, and other blocks. If the SR module can be adjusted easily via our calibration framework, engineers responsible for different pipeline stages can directly specify the desired high-frequency characteristics;
>
> In this way, our method is best viewed as a **complementary tool** that makes large-scale training more targeted and efficient, rather than as a competitor. It can also support practical products features. For example, in a multi-camera setting, a user-facing “detail” slider (e.g., 0–10) built on our correction module would let users choose different image qualities than those produced by a fixed SR model.

---

> ### Author Response · Authors · 2025-11-28
>
> ### **Q3 (Theoretical Maturity).**
>
>
> We acknowledge that there were notational mistakes in the original draft. We thank Reviewer Zmc6 for the careful reading that helped us clarify these points. In our response to Reviewer Zmc6, we clarified that **our assumption do not introduce stronger assumptions than ResShift**. We hope that this clarification also alleviates the present reviewer’s worries about the soundness of our theoretical framework.
>
> At the same time, we would like to emphasize that the *method and experiments themselves* remained unchanged; we changed the way we articulate our main idea.
> 1. Why we chose to use “Proposition” instead of “Theorem”.
>
>    The main reason for this change is to adopt more conservative therminology and avoid potential disagreement over terminology in a short review cycle. Our result collects the existing assumptions of ResShift (Gaussian, isotropic variance per step) and adds our modeling approximation on top. The distinction between "Theorem" and "Proposition" is somewhat subjective in applied work. However, to avoid overstating the claim and to respect the reviewer’s concerns, we chose the more modest label *Proposition* and emphasized that the result is supported experimentally rather than as a fully general theorem.
>
> 2. Perspective on rigor and review.
>
>    We see the review process precisely as a place where such refinements should happen. However, The **underlying theory remained sound”** and The **initial experiments were not altered**;

---

### Official Review · Reviewer_gyNZ · 2025-11-04

**Soundness:** 3
**Presentation:** 3
**Contribution:** 3
**Rating:** 8
**Confidence:** 4

**Summary:**

This paper proposes a plug-and-play framework for posterior diffusion-based image super-resolution (SR), enabling controllable image quality without retraining. The authors reveal discretization errors correspond to KL divergence gradients, linking them to classifier guidance. Their method corrects fidelity degradation using second-order terms and supports both supervised fidelity enhancement and unsupervised perceptual improvement. This work bridges numerical analysis and perceptual conditioning, provides a principled explanation of fidelity degradation and a new lens for posterior diffusion trajectories.

**Strengths:**

1. This paper propose the first plug-and-play module for posterior diffusion-based super-resolution models, which allows pretrained models to perform image quality control without requiring retraining.
2. This paper prove that discretization errors in posterior diffusion can be interpreted as gradients of KL divergence, suggesting that such errors behave like KL gradients between continuous and discrete processes at each sampling step.
3. This paper reveals a theoretical connection between numerical error correction and classifier guidance, and propose an image-based classifier guidance formulation for posterior diffusion, unifying numerical analysis with guidance-based conditioning.
4. Experiments show strong perceptual quality and competitive performance with state-of-the-art models.

**Weaknesses:**

1. The article "Toward real-world single image super-resolution: A new bench mark and a new model" appears twice in the reference list. Similar mistakes exist for several other articles.

2. The most updated methods, SinSR and Osediff, are excluded from visual comparisons. It's not fair.

3. The visual comparisons are insufficient. There's only one scene for reference region and non-reference region, respectively.

**Questions:**

1. In Table 2, the proposed error correction plugin increases all non-reference metrics and most reference-based perceptual metrics but decreases reference-based fidelity measures, especially PSNR. This contradicts the paper's claim that error correction will improve fidelity.

2. In Figure 2, the fidelity of SwinIR outperforms all compared methods. Do you think diffusion models with your proposed error correction technique can possess a similar fidelity as SwinIR in a reference region?

---

> ### Author Response · Authors · 2025-11-26
>
> We thank the reviewer for the constructive comments and address each point below.
>
> **1. Duplicate references in the bibliography**
> We appreciate the reviewer for pointing this out. We carefully reviewed the entire bibliography and removed all duplicated and inconsistent entries in the revised submission.
>
> **2. Missing visual comparison with SinSR and OseDiff**
> Thank you for noting this omission. We have added SinSR, OseDiff, and InvSR to our visual comparison.
> Please refer to Figure 12 in the supplemental material, where both methods are included.
>
> **3. Only one scene shown for the reference and non-reference regions**
> We agree that additional examples improve clarity. We expanded our qualitative comparisons and added multiple additional scenes, including both reference and non-reference regions. Please see Figure 13 in the supplemental material for the updated results.
>
> **4. Error correction improves perceptual metrics but decreases PSNR**
> This is an important point. Our analysis shows that while the numerical correction term theoretically improves fidelity when HR guidance is used, in blind SR (non-reference regions), the LR-based guidance cannot fully align with the HR trajectory. As a result, applying the positive correction improves psnr while the negative correction improves perceptual metrics.
> We clarified this trade-off in Sec. 6.1 and provided additional examples in supplemental Figures 12–13.
>
> **5. Fidelity comparison with SwinIR**
> SwinIR achieves the highest PSNR/SSIM primarily because it is a *deterministic regression model* trained directly with pixel-wise L1/L2 losses. These losses are mathematically aligned with PSNR/SSIM optimization, and therefore SwinIR produces the solution that minimizes per-pixel error with respect to the HR target.
>
> In contrast, diffusion-based SR models—including ResShift, StableSR, DiffBIR, SinSR, and PASD are generative posterior samplers. They emphasize perceptual realism and reconstruct high-frequency textures that may not exactly match the HR ground truth by using perceptual loss. Such perceptual enhancements improve LPIPS/MANIQA/CLIPIQA but are penalized by PSNR/SSIM.
>
> For these reasons, SwinIR remains the top performer in PSNR/SSIM, while diffusion-based SR methods excel in perceptual metrics.
>
> If we have overlooked any part of your question or if you have further concerns, we would be happy to clarify.

---

### Author Response · Authors · 2025-11-26

**Revised Parts in the PDF**

- **Section 2. Related Works**
  - Updated Image Super-Resolution paragraphs.

- **Section 3.3 Theoretical Foundation**
  - Corrected math notations and refined assumptions.
  - Updated *Theorem 3.2.1* → **Proposition 3.2.2**.
  - Added **Proposition 3.2.3**.

- **Section 3.4 Image Control via Image-Based Guidance**
  - Revised mathematical symbols and clarified gradient definitions.

- **Section 4. Experiment**
  - Updated qualitative and quantitative analysis.
  - Several tables moved to the supplementary material.
  - Added Tables **3, 4, 6**. Update dists metric in ours(lr) Table 2

- **Reference Section**
  - Removed redundant citations.

- **Figures**
  - Added **Figure 6** and **Figure 7(b)**.
  - Added additional results:
    - **Figure 12**: Additional comparison with SOTA methods.
    - **Figure 13**: Ablation study.

- **Section 7. Mathematical Details**
  - Unified notations and clarified derivations.

- **Section 9. Future Works**
  - Added a discussion on generality and future posterior-model research directions.

- **Section 10. $\nabla$ $D_{KL}$ as unsharp masking**
  - Added for clarification of sign-flip based correction

---

### Meta-Review · Area_Chair_ARn1 · 2026-01-07

**Summary:**

This paper presents a technically interesting attempt to add plug-in image quality control to posterior diffusion super-resolution, supported by a nontrivial theoretical analysis of discretization errors. Several reviewers appreciated the new perspective and the effort to connect numerical analysis with practical controllability. However, concerns remained about the strength of the underlying assumptions, the limited empirical gains over existing posterior SR methods, and the robustness of calibration in real-world settings. Overall, the work is promising and clearly improved through rebuttal, but it is not yet strong enough for acceptance at this time.

**Reviewer Concerns:**

The rebuttal addressed many concrete concerns, including notation errors, theoretical overstatements, missing comparisons, calibration stability, and qualitative analysis, which improved the paper’s clarity and rigor. Reviewers gyNZ and ReCH’s concerns were largely resolved, and Zmc6 explicitly acknowledged improvement and raised their score. However, concerns from zA8j and Zmc6 about the conceptual leap behind sign-flipping, calibration robustness, strong Gaussian assumptions, and limited practical gains remain only partially resolved.

**Reviewer Scores:**

Reviewer gyNZ would likely keep a strong accept, since the presentation and experimental issues were largely fixed. Reviewer ReCH would probably stay at a marginal accept, as the calibration limitations are now clearer but still present. Reviewer Zmc6 already raised the score from 2 to around 4 after the rebuttal, but remains unconvinced by the core theoretical assumptions and is unlikely to move further. Reviewer zA8j was also not fully persuaded during the discussion, and would likely remain borderline or slightly negative due to ongoing concerns about robustness, calibration, and conceptual soundness. The scores after rebuttal could remain mixed, with a slight lean toward the negative side.

---

### Decision · Program_Chairs · 2026-01-26

Reject